applied mathematics/complexity/graph theory

directed network, trophic level, trophic coherence

**Author for correspondence:**
R. S. MacKay
e-mail: r.s.mackay@warwick.ac.uk

# How directed is a directed network?

R. S. MacKay[1,4], S. Johnson[3,4] and B. Sansom[2]

[1]Mathematics Institute and Centre for Complexity Science, and [2]Mathematics and Economics, University of Warwick, Coventry, UK
[3]School of Mathematics, University of Birmingham, Birmingham, UK
[4]The Alan Turing Institute, London, UK

RSM, 0000-0003-4771-3692; SJ, 0000-0002-8648-1735; BS, 0000-0003-1701-8453

The trophic levels of nodes in directed networks can reveal their functional properties. Moreover, the trophic coherence of a network, defined in terms of trophic levels, is related to properties such as cycle structure, stability and percolation. The standard definition of trophic levels, however, borrowed from ecology, suffers from drawbacks such as requiring basal nodes, which limit its applicability. Here we propose simple improved definitions of trophic levels and coherence that can be computed on any directed network. We demonstrate how the method can identify node function in examples including ecosystems, supply chain networks, gene expression and global language networks. We also explore how trophic levels and coherence relate to other topological properties, such as non-normality and cycle structure, and show that our method reveals the extent to which the edges in a directed network are aligned in a global direction.

## 1. Introduction

Many complex systems have an underlying network, whose nodes represent units of the system and whose edges indicate connections between the units [1]. In some contexts, the connections are symmetric, but in many they are directed, for example, indicating flows from one unit to another or which units affect which other units [2]. A classic example is a food web, in which the nodes represent species and there is a directed edge from each species to those which eat it.

In a directed network, the ecological concept of 'trophic level' [3] allows one to assign a height to each node in such a way that on average the height goes up by one along each edge. The trophic levels can help to associate function to nodes, for example, plant, herbivore, carnivore in a food web. The concept was reinvented in economics [4], where it is called 'upstreamness', though [5] trace it back to Leontief and the 'output multiplier'. It is also an ingredient in the construction of SinkRank, a measure of contribution to systemic risk [6].

The standard deviation of the distribution of height differences along edges gives a measure of the extent to which the directed edges fail to line up, called the 'trophic incoherence' [7]. The trophic incoherence is an indicator of network structure that has been related to stability, percolation, cycles, normality and various other system properties [8–12].

The standard definitions of trophic level and incoherence are limited in various ways, however. In particular, they require the network to have a basal node (a node with no incoming edges), they give too much emphasis to basal nodes if there is more than one, they do not give a stable way to determine levels and incoherence for a piece of a network, and they do not give a natural notion of maximal incoherence. Furthermore, in some contexts, like production networks indicating the flows of goods and services between firms or sectors, the reverse flow plays an equivalent role, representing the financial payments, but the standard concept of trophic level does not treat these symmetrically.

In this paper, we present improved[1] definitions of trophic level and incoherence that overcome these limitations. We illustrate their application in a variety of domains. We show that the new levels continue to be a useful indicator of function in the network and that the new incoherence measure continues to be related to stability, cycles and normality. We compare the new notions with the old for cases that have basal nodes; and we show the robustness of our new trophic levels to truncation of a network. Mathematical proofs are given in appendices.

## 2. The improved notions of trophic level and incoherence

We consider directed networks (also known as directed graphs or digraphs) with set $N$ of nodes (also known as vertices) and set $E$ of directed edges (also known as links or ties). We suppose that there is at most one edge from a node $m$ to a node $n$, and denote the edge by $mn$. There can also be an edge from $n$ to $m$. Each edge carries a weight $w_{mn} > 0$. This can represent the strength of the edge, for example the amount of flow along it or a quantification of influence of one node on another. We write $w_{mn} = 0$ if there is no edge from $m$ to $n$ and we assemble the $w_{mn}$ into a matrix $W$. The edge weights could be set to 1, as is common in the literature, and the array $W$ is then called the adjacency matrix $A$ of the network, but the ability to represent the strength of the edge is a useful extension. If there were multiple edges from $m$ to $n$, then we would amalgamate them into a single edge by adding the weights. Self-edges $mm$ (also called loops) are permitted.

For each node $n$, we define its in-weight and out-weight by

$$w_n^{\text{in}} = \sum_{m \in N} w_{mn} \quad \text{and} \quad w_n^{\text{out}} = \sum_{m \in N} w_{nm}. \tag{2.1}$$

Alternative terminology could be in- and out-strength, extending [13], who called the out-weight of a node its *strength*. We define the *(total) weight* of the node $n$ by

$$u_n = w_n^{\text{in}} + w_n^{\text{out}}, \tag{2.2}$$

and the *imbalance* for node $n$ by

$$v_n = w_n^{\text{in}} - w_n^{\text{out}}, \tag{2.3}$$

the latter representing the difference between the flow into and out of the node. We make vectors $u$ and $v$ from the $u_n$ and $v_n$. The (weighted) *graph-Laplacian operator* $\Lambda$ on vectors $h$ is defined by

$$(\Lambda h)_m = u_m h_m - \sum_{n \in N} (w_{mn} + w_{nm}) h_n, \tag{2.4}$$

or in matrix form (where superscript T denotes transpose),

$$\Lambda = \text{diag}(u) - W - W^{\text{T}}. \tag{2.5}$$

Then our improved notion of *trophic level* is the solution $h$ of the linear system of equations

$$\Lambda h = v, \tag{2.6}$$

---

[1]Although we use the words 'improved' and 'new', it is up to the reader to assess if you agree they are improvements, and after writing the first version we found precursors of our definitions, to be reported in §4; nevertheless, our analyses go significantly beyond them, particularly in the quantification of trophic incoherence and its connection with other network properties.

modulo shifts to be characterized in the next paragraph. Note that although the operator $\Lambda$ is symmetric, asymmetry of the network appears in the imbalance vector $v$. Comparisons with previous notions will be made in §4.

Equations (2.6) always have a solution (see appendix A.1) but it is non-unique, because one can add an arbitrary constant in each connected component of the network. A *connected component* (more correctly called 'weakly connected component') of a network is a maximal subset $S \subset N$ such that it is possible to get from any $m \in S$ to any $n \in S$ by a path of edges ignoring their directions. Thus to solve $\Lambda h = v$, one can replace the equation for one node $m_S$ in each connected component $S$ by an equation $h_{m_S} = c_S$ for arbitrary constants $c_S$, for example 0. Then there is a unique solution for $h$, which can be found by any linear algebra package. Afterwards one can add an arbitrary constant to the levels in each component $S$ if desired, for example to make the lowest one be 0 or to make the average level (with respect to the weights $u_n$, for example) in $S$ be 0.

Our improved notion of *trophic incoherence* is

$$F_0 = \frac{\sum_{mn} w_{mn}(h_n - h_m - 1)^2}{\sum_{mn} w_{mn}}, \tag{2.7}$$

using the levels $h$ determined above; it is independent of the choice of shifts on connected components. This has the nice features that $F_0 = 0$ if and only if all the level differences $z_{mn} = h_n - h_m$ are 1, $F_0 = 1$ if and only if all the level differences are 0, and otherwise $F_0$ is strictly between 0 and 1 (see appendix A.2 for a proof). We say a network is *maximally coherent* if it has $F_0 = 0$, *maximally incoherent* if it has $F_0 = 1$. We define the trophic coherence to be $1 - F_0$. In appendix A.3 we prove the trophic coherence can be expressed alternatively as the weighted mean difference $\bar{z}$ in trophic levels between nodes along the edges of the network.

The motivation for our new definitions is to seek levels $h_n$, $n \in N$, that minimize the trophic confusion

$$F(h) = \frac{\sum_{mn} w_{mn}(h_n - h_m - 1)^2}{\sum_{mn} w_{mn}}, \tag{2.8}$$

where the target level difference for each edge $mn$ is set to 1. A vector $h$ of levels minimizes $F$ if and only if $\Lambda h = v$ (see appendix A.2). The resulting minimum value of $F$ is the incoherence $F_0$.

The coherence $1 - F_0$ represents the extent to which levels can be assigned to make the level difference along each edge be the target difference of 1. So we can alternatively call $1 - F_0$ the 'directionality' or 'directedness' of the network, hence the title of our paper.

For an electrical interpretation of (2.6), see appendix A.4.

# 3. Illustrations

To illustrate the new notions of trophic level and incoherence, we begin with the classic context of food webs. Here the nodes represent species and there is a directed edge from a species to each species that eats it. Figure 1 shows the Ythan estuary food web [14] with height in the layout corresponding to our new notion of trophic level. The networks are spread out in the horizontal dimension by a force-based algorithm (details and refinements will be presented in a future publication). The network is fairly strongly layered; this is borne out by a small value of trophic incoherence $F_0 = 0.08$.

We move now to an example from economics where the 'upstreamness'/'downstreamness' of *firms*, *sectors* and *economies* in production chains is of wide relevance and interest [5,15,16]. Figure 2 shows the inter-industrial flows of goods and services in the USA and Saudi economies in 2015 (data taken from OECD IO tables). Here the nodes represent economic sectors and weighted edges represent the dollar value of supply → purchase transactions between them (the full IO table had 35 sectors, but nodes with lower weight (2.2) were removed to allow presentation of a labelled network). This is an interesting application because there are no basal nodes (indeed the networks are completely connected, as is usual for IO relations, with every sector both supplying and buying from every other sector), so the old notions of trophic level and incoherence cannot be applied.

Unlike the Ythan food web, we see that these IO networks are rather incoherent, meaning their trophic incoherences are not small ($F_0 = 0.63$ and $F_0 = 0.46$, respectively). Nevertheless, the new levels reveal an overall direction between sectors of flow in intermediate production: some sectors are key suppliers of intermediate inputs (for the USA, financial, real estate and other business service sectors; for Saudi Arabia, energy extraction and finance) while other sectors are key users of inputs from other sectors (e.g. healthcare and construction). Some sectors, however, are both users and suppliers of products (e.g. wholesale, transport and storage).

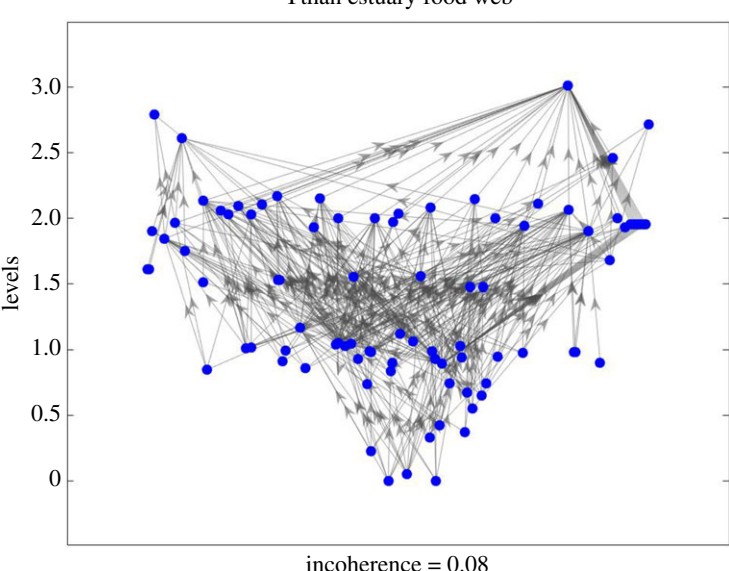

**Figure 1.** Ythan estuary food web with height corresponding to our new trophic levels which reveal a strongly layered structure. Edges represent *prey → predator* relations and edge weights are all taken to be 1 as the data do not specify the relative importances of relationships.

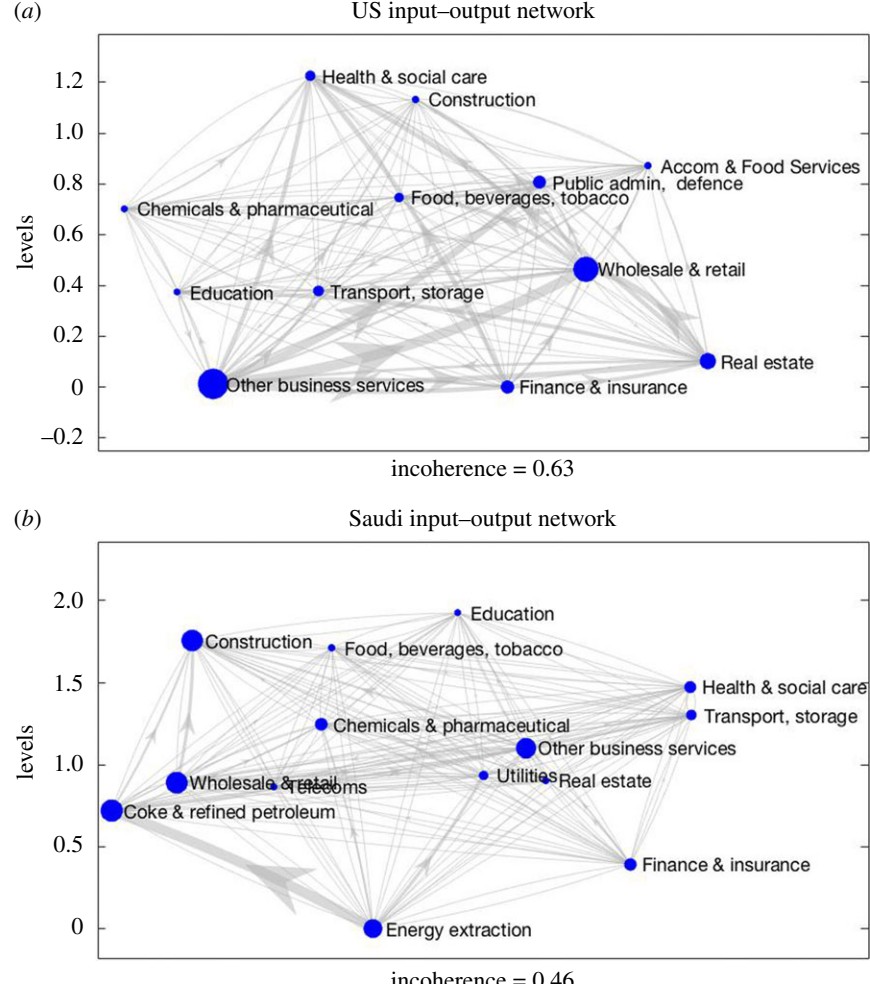

**Figure 2.** Network of inter-industrial flows of goods and services in the USA (*a*) and Saudi (*b*) economies in 2015. Nodes represent a subset of economic sectors (accounting for largest share of inter-industry flows as captured by weight (2.2)) and weighted edges represent the dollar value of *supply → purchase* transactions between them. Edge widths reflect the value of flows, and node size reflects node weight (2.2).

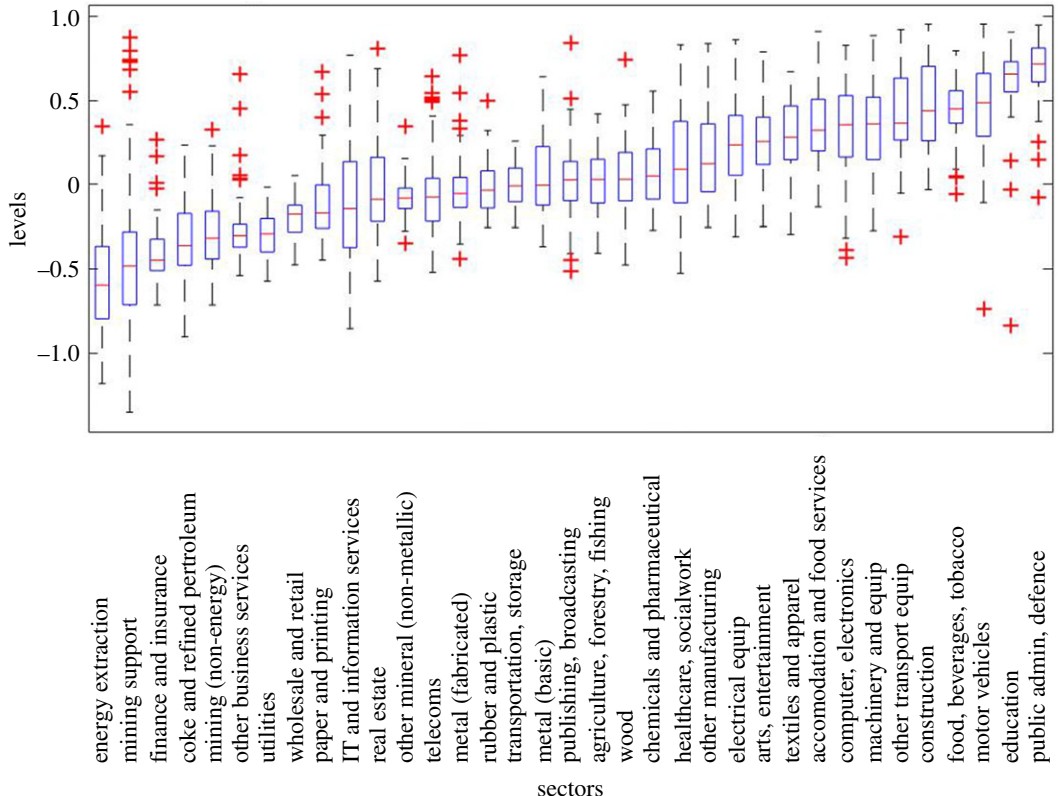

trophic levels of different sectors in IO nets for 57 economies

**Figure 3.** These boxplots present the distribution of the new trophic levels for each of 35 different economic sectors (ISIC Rev. 4) as obtained from the national 2015 input–output (IO) networks of 57 different economies (including OECD, and G20 economies). Sectors are sorted by their median trophic level (across all 57 IO networks). Red crosses indicate outliers.

Figure 3 provides a more systematic and detailed analysis, presenting box-plots of the level of different sectors (using full 35 sector IO tables) for 57 countries (2015 data). Levels for each economy have been normalized to make the mean level 0 (weighted by $u_n$). While the size of different sectors varies across economies, there is considerable consistency of sector levels, which reveal the architecture of value chains in the production process: we see an overall direction of flow from energy extraction and finance sectors; through other primary materials; then manufacturing industries; followed by sectors that supply final demand more directly, such as food makers, entertainment and services; ending with education, public administration and defence sectors (that are overwhelmingly users more than suppliers of intermediate inputs).

There may be links to explore between sector levels and their role in economic performance—it is interesting to note, for example, that construction—which is known to lead the wider business cycle in many economies [17,18]—appears as a key user of inputs from other industries (implying strong backward-linkages). Meanwhile variation in the level of some sectors across different economies may also reveal interesting differences in production structure (e.g. finance occupies the same minimum position as energy extraction in China, but comes higher in the value chain for many other economies).

In biology, regulatory networks are sets of macromolecules that interact to control the level of expression of various genes in a given genome [Nature subjects: Regulatory networks]. Studies on regulatory networks have identified the existence of directed structures and have linked node-levels to node-properties, function [19–21] and the importance of regulators [22]. Assigning levels in networks with cycles, however, has presented a methodological challenge for this literature, which our improved levels overcome. Figure 4 shows an example transcription regulatory network (the yeast *Saccharomyces cerevisiae* [20]) plotted first with a force-directed method (left), then according to the new levels (right). The new levels reveal a striking layered structure. There are basal (red), intermediate (yellow) and top (blue) nodes, but intermediate nodes do not form a distinct layer and the relevance of variation in their levels might be explored.

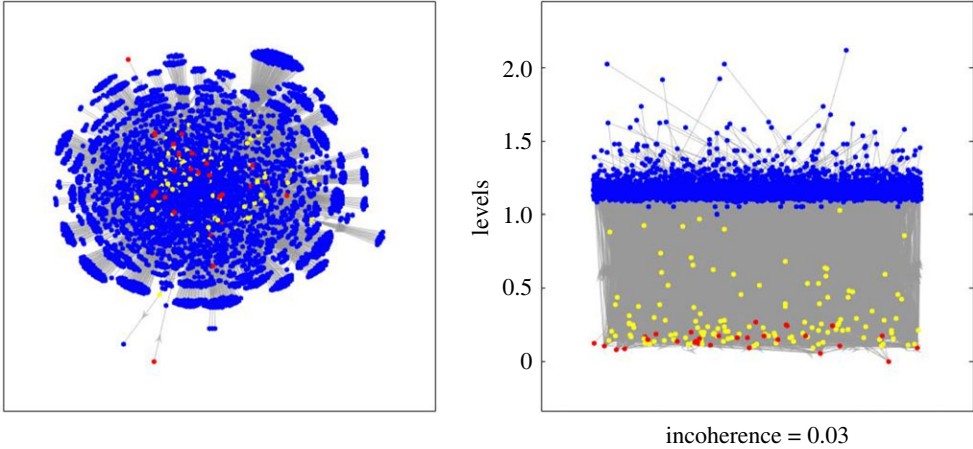

**Figure 4.** These two charts plot the same yeast transcription regulatory network (linking transcription factors and target genes) first with a standard force directed layout (left); then with node heights corresponding to new trophic levels $h$ to reveal the network's flow-based hierarchy. Red nodes represent transcription factors, blue nodes denote regulated genes, and those with both functions are coloured in yellow.

Levels derived from influence networks will also be useful in social network settings (hierarchy and stratification are important concepts in sociology) and have been studied in e.g. online social networks [23,24].

Figure 5 shows the trophic analysis of a network of book translations [25] based on a collection of more than 2.2 million book translations compiled by UNESCO's *Index Translationum project* [26]. Edge weights correspond to the number of books translated between source and target languages. While it is unlikely that individual books flow along paths in this network (given books are presumably translated from their original source language) its structure may be important in the flow of knowledge and ideas [25].

The role of eigenvector centrality in the influence of different languages within this network was studied in [25]. Our trophic analysis reveals that this network is strongly directional ($F_0 = 0.51$), implying knowledge and ideas are not exchanged equally but flow in particular directions. It also reveals interesting information on the role of different languages within this directional flow: at the bottom appear languages that are only source languages—unsurprisingly these include many 'dead languages' (Ancient Greek, Middle French and English, Sanskrit, etc.). At the top appear languages from which nobody translates (these include minority and other languages that are small by number of speakers such as Faroese, Sami and Mongolian). In the middle, we find languages that are both target and source languages. The role of English is striking: while translated into and out of, English is more important as a *source* language (it has a lower trophic level than any other major language) and there are large flows from English into French, German, Spanish and Japanese. In this dataset only English is translated into Chinese which is in turn only a source language for minority languages in China (such as Hani and Zhuang). Russian is rather isolated in the global language network but forms an interesting community of bi-directional links with languages in its region.

## 4. Comparison with previous notions

The established concept of trophic level [3] requires the network to have at least one basal node, i.e. a node with no incoming edges. Then the height $x_b$ (to use Levine's symbol) was set to a common value of 0 for all basal nodes $b$, though nowadays it is more common to set it to 1. The heights $x_n$ of the other nodes in connected components with basal nodes were determined by solving

$$x_n = 1 + \frac{\sum_m x_m w_{mn}}{w_n^{in}}, \tag{4.1}$$

for all non-basal $n$, where each sum is over the nodes $m$ having edges to $n$. Levine normalized the weights $w_{mn}$ coming into each node $n$ so that $w_n^{in} = 1$, which makes no change to (4.1). In matrix form, the

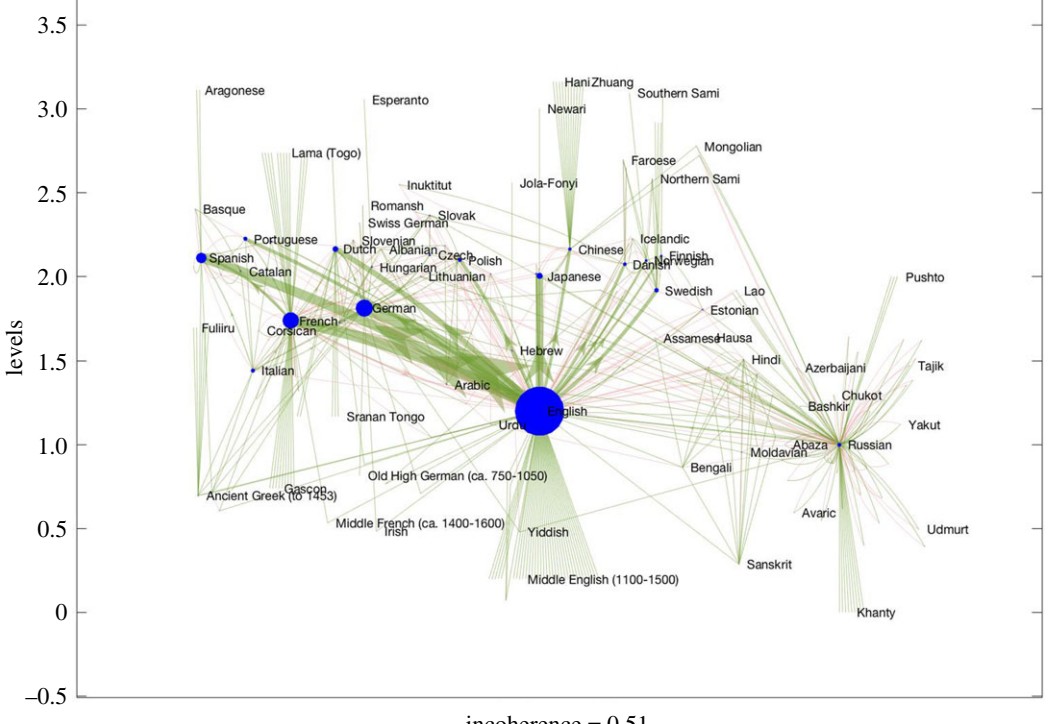

incoherence = 0.51

**Figure 5.** Global book translation network [25]. Edges and edge weights represent the number of books translated from source into target language. Upward arrows are plotted green and downward arrows red. Node size is proportional to weight (2.2).

equation for the heights (with the convention $x_n = 1$ for basal nodes) can be written as

$$Lx = \tilde{v}, \tag{4.2}$$

where

$$\tilde{v}_n = w_n^{\text{in}} \quad \text{if non-zero, else 1,} \tag{4.3}$$

and

$$(Lx)_n = \tilde{v}_n x_n - \sum_m x_m w_{mn}. \tag{4.4}$$

The same concept was introduced in economics by Antràs *et al.* [4], but fixing top nodes (those with no outgoing edges) to a common height. It is equivalent to Levine's after reversing all the edges.

Then [7] defined the trophic incoherence of the network to be the standard deviation of the height differences $z_{mn} = x_n - x_m$ over edges. They took edge weights all 1, but a natural generalization is to weight the height differences by the edge weights. The edge-weighted mean difference of Levine's heights is precisely 1 [3], so Johnson *et al.*'s [7] definition of trophic incoherence $q$ becomes

$$q = \sqrt{\frac{\sum_{mn} w_{mn}(x_n - x_m - 1)^2}{\sum_{mn} w_{mn}}}. \tag{4.5}$$

Indeed, Levine defined 'trophic specialization' of a node $m$ as

$$\sigma_m^2 = \frac{\sum_n w_{mn}(x_n - x_m - 1)^2}{\sum_n w_{mn}}. \tag{4.6}$$

So $q^2$ is the average of $\sigma_m^2$ weighted by $w_m^{\text{out}}$.

Our equation for trophic heights can be seen as a symmetrized version of Levine's, without the fix for basal nodes. Thus, our definition does not need any basal nodes and does not force them all to the same level if there is more than one basal node.

Our definition of trophic incoherence is the same as $q^2$ but using our new heights instead of Levine's. It represents, in roughly the same way, the failure of the height differences to all be 1. A distinction to bear

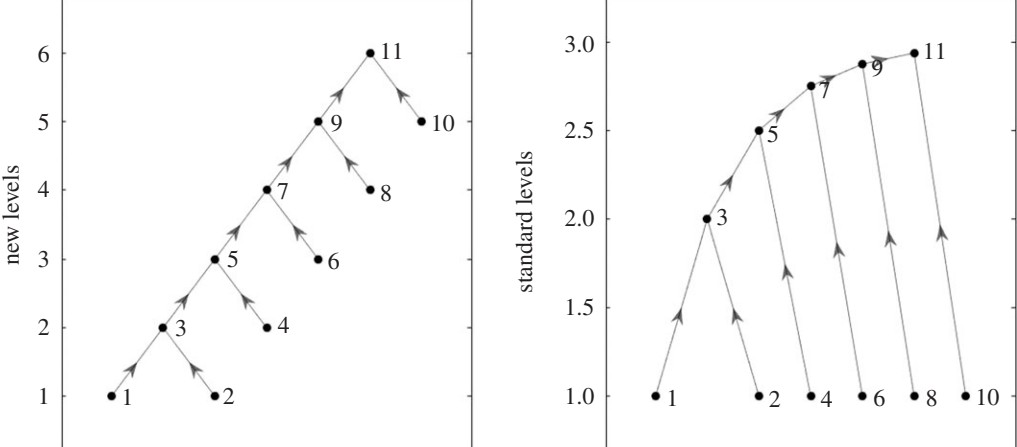

**Figure 6.** This figure shows a comparison between our levels versus standard levels calculated for a toy example of a sequential process with six stages.

in mind, however, is that for our new levels, the edge-weighted mean height-difference

$$\bar{z} = \frac{\sum_{mn} w_{mn}(h_n - h_m)}{\sum_{mn} w_{mn}}, \tag{4.7}$$

is not necessarily 1. In fact, we prove in appendix A.3 that $\bar{z} = 1 - F_0$. So $F_0$ is not in general the (edge-weighted) variance of the height differences. To obtain the variance $\sigma^2$ of the height differences, one has to subtract $(\bar{z} - 1)^2$ from $F_0$. Thus there is a case for considering alternative measures of incoherence to $F_0$, such as the ratio $\eta = \sigma/\bar{z}$, which evaluates to

$$\eta = \sqrt{\frac{F_0}{1 - F_0}}, \tag{4.8}$$

and is the appropriate replacement for $q$. In the other direction, the analogue of $F_0$ is $q^2/(1 + q^2)$.

While we acknowledge there may be some applications where fixing all basal (or top) nodes to the same level may be most appropriate, we believe that in many cases it will be more natural to allow the height of basal nodes to be determined according to their integration into the overall flow hierarchy of the network. If we take, for example, figure 6 as representing a simple stylized sequential production process where intermediate inputs are transformed over six stages, it seems to us that this process is better described by levels according to our notion—which returns integer levels identifying the discrete stages of the process and which stage each node belongs to—than it is by standard levels. It seems natural to consider node 10 to be more upstream than node 1 or 2 (although they are all 'basal'), and this may be important in the context of e.g. sector specific shocks.

Of course the flow hierarchy of most real processes will be a more complex web of intermediate flows. Figure 7 shows some comparisons of trophic levels determined by the two methods for two different empirical supply networks. In these networks, the nodes represent firms and each directed edge represents a significant *supplier → buyer* relationship.[2] We see that the requirement of the standard approach to put all basal nodes at a common level makes what we consider to be an artificial distortion of the levels—especially in the left-hand case.

As an alternative comparison, in figure 8 we plot (for the same two supply networks as figure 7) the old levels against the new levels.

If one reverses all the edges, then with our new definition one obtains the reflection of the trophic levels, up to an overall shift depending on the convention used to fix the zero of the levels. The trophic incoherence is unchanged. For example, for a supply network, instead of the flows of goods and services one could consider the flows of payment, which are more or less the reverses of the flows of goods and services.

By contrast, the old notion of trophic level is usually not symmetric with respect to change of direction of all the edges. Figure 9 shows the trophic levels of firms in our two example supply

---

[2]These networks were extracted from Bloomberg by taking all suppliers and buyers within 3 hops (a hop being an edge in either direction) of a given firm (in this case, Lockheed Martin Corp. and the Tractor Supply Company).

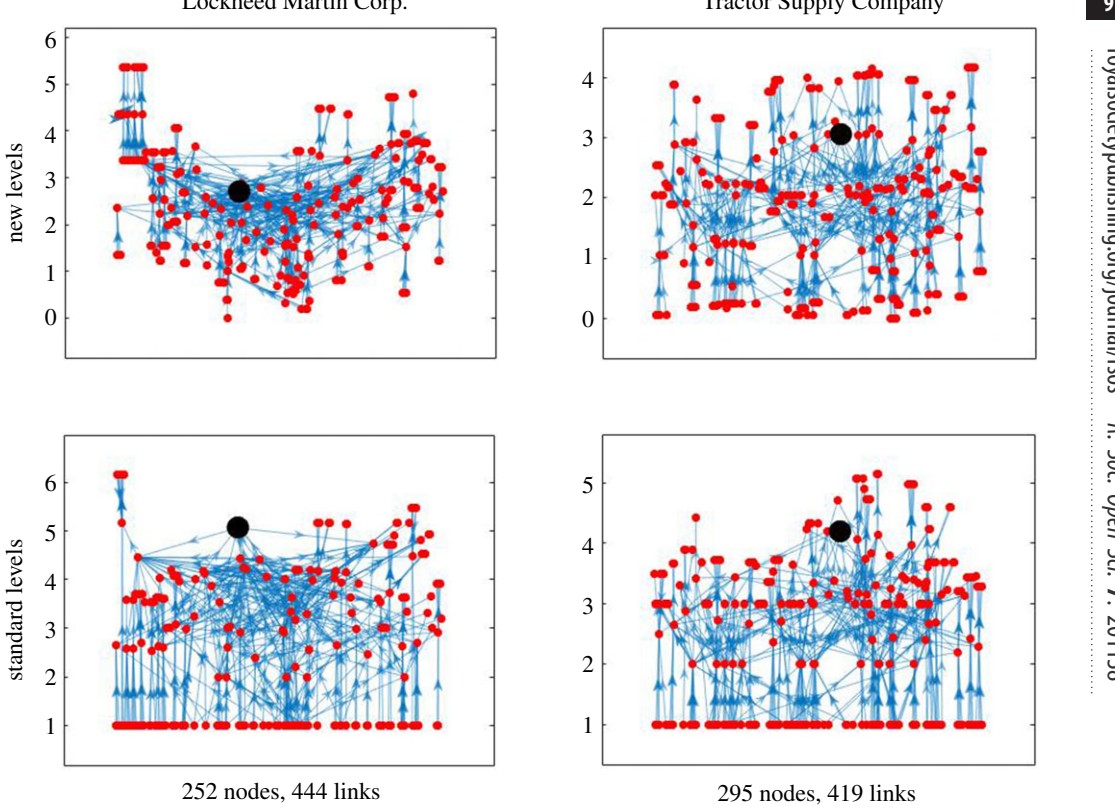

**Figure 7.** Supply networks around two firms (indicated by the larger black dot), plotted with the new (top) and old (bottom) notions of trophic level. The horizontal positions are determined to spread out the nodes while attempting to make most of the edges near vertical, but the same horizontal positions are used in both the upper and lower pictures. Supply chain data compiled from Bloomberg L.P.

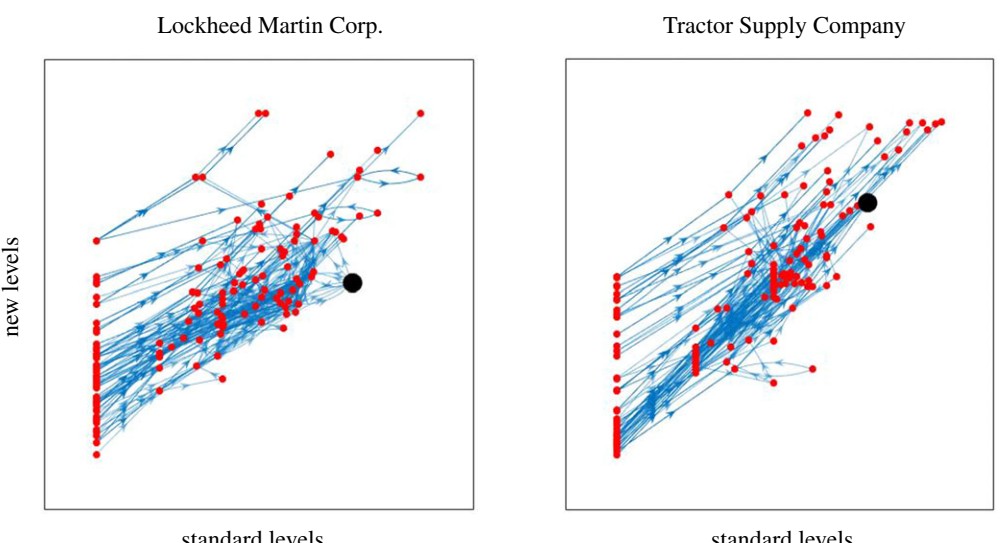

**Figure 8.** New levels against old levels for the same two supply networks as in figure 7.

networks obtained according to the old notion, (i) when edges are directed from supplier to buyer (showing the direction of material and service flows), and (ii) under the reverse interpretation (showing the direction of payment flows from buyers to sellers). It is apparent that with the old notion there is a big change in levels, the relevance of which is unclear. Unless there is a good reason to favour basal nodes, we propose that our symmetric notion is advantageous.

Lockheed Martin Corp.                Tractor Supply Company

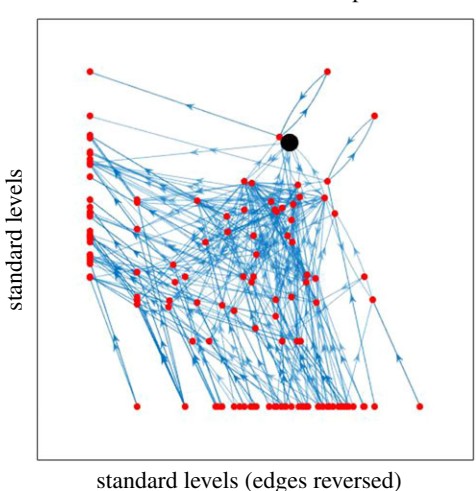 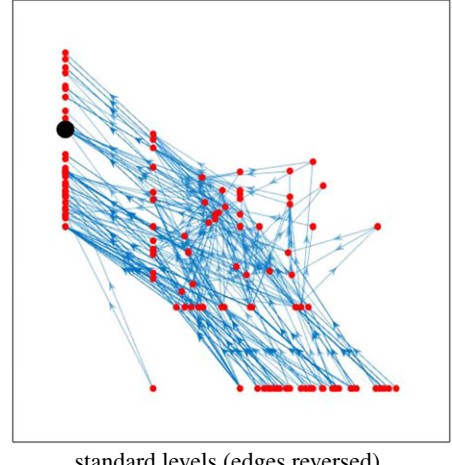

standard levels (edges reversed)        standard levels (edges reversed)

**Figure 9.** The same two supply networks as in figures 7 and 8 with nodes organized according to trophic levels obtained using the old notion for (i) the original networks (vertical axis) and (ii) the same networks but with interpretation of edges reversed (horizontal axis).

There have been some other approaches to rectifying the limitations of the original notion of trophic level. Dominguez *et al.* [8] obtain a 'basal set' of nodes and eliminate all edges within that set. Moutsinas *et al.* [27] define levels using a pseudo-inverse of $L$. These solutions allow application to networks without basal nodes but they do not possess a natural notion of maximal incoherence nor symmetry with respect to reversal of edge directions (though in some contexts one might not want symmetry).

Trophic incoherence is related to the number of directed cycles in a network [10], and hence to measures of acyclicity. For example, one can find the smallest number of edges to delete to obtain an acyclic graph [28], although this method has some defects [29]. The smallest number is called the 'agony' of the network. Our trophic analysis provides a useful upper bound for agony, given by the number of edges with negative height difference, and could be used as a convenient heuristic for its exact computation. However, agony and incoherence are different concepts, since acyclic networks can differ in incoherence [7]. Related papers are [30] on reordering a matrix to make it as triangular as possible, and [31] quantifying directedness by the fraction of edges not in any cycle.

A precursor of our notion of trophic levels was given by [32], with the same minimization principle (extended to arbitrary height differences as we do in §7), but they chose a different quantification of directedness, namely the ratio of the difference in levels of the highest and lowest nodes to the diameter of the network, which is more sensitive to extremes and in our opinion less correlated with the other network properties we have considered. The same minimization principle was proposed again by De Bacco *et al.* [33], but without a quantification of directedness. The recent papers [34,35] derive essentially the same notions of levels and coherence as us, by decomposing flows on a network into the sum of a potential part and a circulating part. This is a very nice approach, though it requires specifying conductivities for each edge as well as the flow on it, instead of specifying a weight and a target height difference for each edge. The analysis has strong connections with ours, in particular the minimization principle to determine the potential and an electrical interpretation. The connections are described in appendix A.14.

Next, we comment on the large literature on hierarchical methods for directed networks. Although the term 'hierarchy' is often used just to describe a node as being above or below another (cf. our use of 'flow hierarchy'), it generally has more connotations that we are not addressing here. In particular, in an ideal hierarchy the graph of immediate superiority relations is often assumed to have a unique maximal node and to form a pyramid, and the question posed is how far a network deviates from this structure. The question overlaps with the one we address, but is different. A full survey of the literature will have to wait for a future publication, but we highlight the book [36] and the following references. Corominas-Murtra *et al.* [37] propose three measures of hierarchy: treeness, feed-forwardness and orderability. Ravasz *et al.* [38] consider containment hierarchy. Ruths & Ruths [39] consider control of directed networks. Czègel & Palla [40] introduce a method to distinguish between

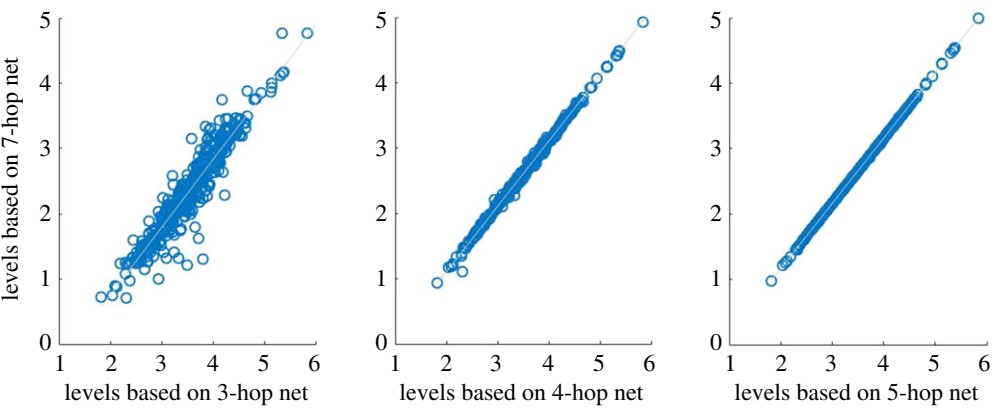

**Figure 10.** Plot of the new trophic levels of buyers and suppliers in a 2-hop neighbourhood of General Motors calculated on networks constructed by sampling neighbourhoods of increasing size (3,4 and 5 hops) (horizontal axes), versus the levels of the same set of nodes calculated on a larger 7-hop neighbourhood (vertical axes). Supply chain information compiled from Bloomberg L.P.

a directed acyclic graph and ones with maximal directedness, something which our method also does. Mones *et al*. [41] introduce a global reaching centrality measure. Maktoubian *et al*. [42] propose how to take signed edge-weights into account.

There are many other concepts of importance for nodes in a network, e.g. PageRank, HITS and Katz centrality (for a survey, see [43]). To the best of our knowledge, however, these are all very different from trophic levels.

## 5. Robustness of local computation

If we determine trophic levels on a piece of a network by truncating the network at some distance from a chosen node, measured for example by the number of edges in either direction, how robust are the resulting levels to the truncation? This is an important question in practice, because it might be infeasible to obtain or analyse the whole of a large network, yet it can be useful to determine relative levels on a piece of the network.

First we take care of the arbitrariness of the zero of trophic levels. The simplest way to do that is to take the chosen node to be always at height zero.

Next, we refine the question because the trophic levels near the boundary of the piece of the network may change significantly with the truncation. We ask how much the trophic levels change on a connected subset of the network containing the chosen node, which we will call zone 1, given a buffer zone 2 chosen so that there are no direct edges in either direction between zone 1 and the outside, called zone 3. We choose the buffer zone so that in addition the union of zones 1 and 2 is connected (the only way this cannot be satisfied is if zone 2 contains nodes which are not connected to zone 1 by a path in zone 2, in which case one can just throw them out).

Figure 10 shows the outcome of a test, taking zone 1 to be the set of suppliers and buyers of General Motors (GM) 2 hops from GM, and computing the effects on the trophic levels in zone 1 of truncation of the network at 3, 4 and 5 hops, respectively (i.e. allowing a zone 2 buffer), compared to truncating at 7 hops. One can see that the trophic levels on zone 1 stabilize quite rapidly.

In appendix A.5, we give some theoretical analysis to support the general conclusion that the levels on zone 1 are robust to changes on zone 3.

## 6. Connections to other network properties

A large part of the interest of the original notion of trophic coherence was its relation to network properties such as the stability of equilibria of Lotka–Volterra dynamics on the network [7], the dynamics of spreading processes [11], prevalence of cycles [10], other motifs [12], intervality [8] and normality [9]. We show here that the new notion of trophic coherence has similar connections, even

stronger, and it enlarges the scope of application because it does not require basal nodes. We examine three of the properties.

## 6.1. Normality

A directed network is said to be *normal* [44] if its weight matrix $W$ commutes with its transpose $W^T$,

$$WW^T = W^TW. \tag{6.1}$$

Note that $W^T$ represents the same weighted network but with all the edges reversed. Empirical directed networks are often highly non-normal [45], so the use of the word 'normal' is somewhat unfortunate in this context.

For the unweighted case of an adjacency matrix $A$, normality implies that the imbalance vector $v$ is 0. This is because $(A^TA)_{mn}$ is the number of sources in common to nodes $m$ and $n$, and $(AA^T)_{mn}$ is the number of targets in common. In particular, $(A^TA)_{nn} = w_n^{in}$ and $(AA^T)_{nn} = w_n^{out}$, so $A^TA = AA^T$ implies that $w^{in} = w^{out}$ and $v = 0$.

When $v = 0$ we say a network is *balanced*. In appendix A.6, we prove that a network is balanced if and only if its trophic incoherence $F_0 = 1$. So, from the previous paragraph, normal unweighted networks are maximally incoherent.

Another special case of normality is symmetric networks $W = W^T$. If $W$ is symmetric then the imbalance vector $v = 0$. So symmetry implies maximal incoherence.

The concept of normality is broader than either of these, however, and maximal incoherence is not equivalent to normality. There are non-normal networks with $v = 0$ and hence maximal incoherence, e.g.

$$W = \begin{bmatrix} 1 & 1 & 0 \\ 0 & 0 & 1 \\ 1 & 0 & 0 \end{bmatrix}. \tag{6.2}$$

Nevertheless, the extent to which a network is normal seems to be positively correlated with its trophic incoherence $F_0$. The degree of normality of a network can be quantified by

$$\nu = \frac{\sum_j |\lambda_j|^2}{\|W\|_F^2}, \tag{6.3}$$

where $\|W\|_F = \sqrt{\sum_{mn} |w_{mn}|^2}$ is called the Frobenius norm of $W$, and $\lambda_j \in \mathbb{C}$ are the eigenvalues of $W$ (with multiplicity). Some of the literature uses $\sqrt{\|W\|_F^2 - \sum_j |\lambda_j|^2}$ as a quantifier of non-normality, but we consider it simpler to use the normality $\nu$ (as in the real elliptic Ginibre ensemble [46]). The normality $\nu$ of $W$ lies in the interval [0, 1], with $\nu = 1$ if and only if $W$ is normal [44]. If $W$ is maximally coherent ($F_0 = 0$) then all its eigenvalues are 0 (appendix A.7), so $\nu = 0$ and it is maximally non-normal. But one can have $\nu = 0$ without $F_0 = 0$, for example, the feed-forward motif (figure 11) with

$$W = \begin{bmatrix} 0 & 1 & 1 \\ 0 & 0 & 1 \\ 0 & 0 & 0 \end{bmatrix}, \tag{6.4}$$

for which $h = [-2/3\ 0\ 2/3]^T$ and $F_0 = 1/9$.

Figure 12 shows normality against trophic incoherence for some real networks. We see that normality increases with $F_0$, but not linearly. In appendix A.10, we present heuristic arguments in favour of a relationship between them of the form $\nu \approx \exp(1 - 1/F_0)$. This is consistent with a relationship between normality and the old notion of trophic coherence [9].

## 6.2. Stability

Next we discuss how dynamical processes on networks are affected by their trophic coherence.

A simple dynamical model for contagion on a weighted network in discrete time is

$$x_n' = \frac{\sum_m x_m w_{mn}}{r}, \tag{6.5}$$

where $x_n \geq 0$ represents the amount of infection at node $n$ at some time, $x_n'$ the amount at the subsequent time, and $r > 0$ is a scaling factor. We wish to know whether the total infection $\|x\|_1 = \sum_n x_n$ on the

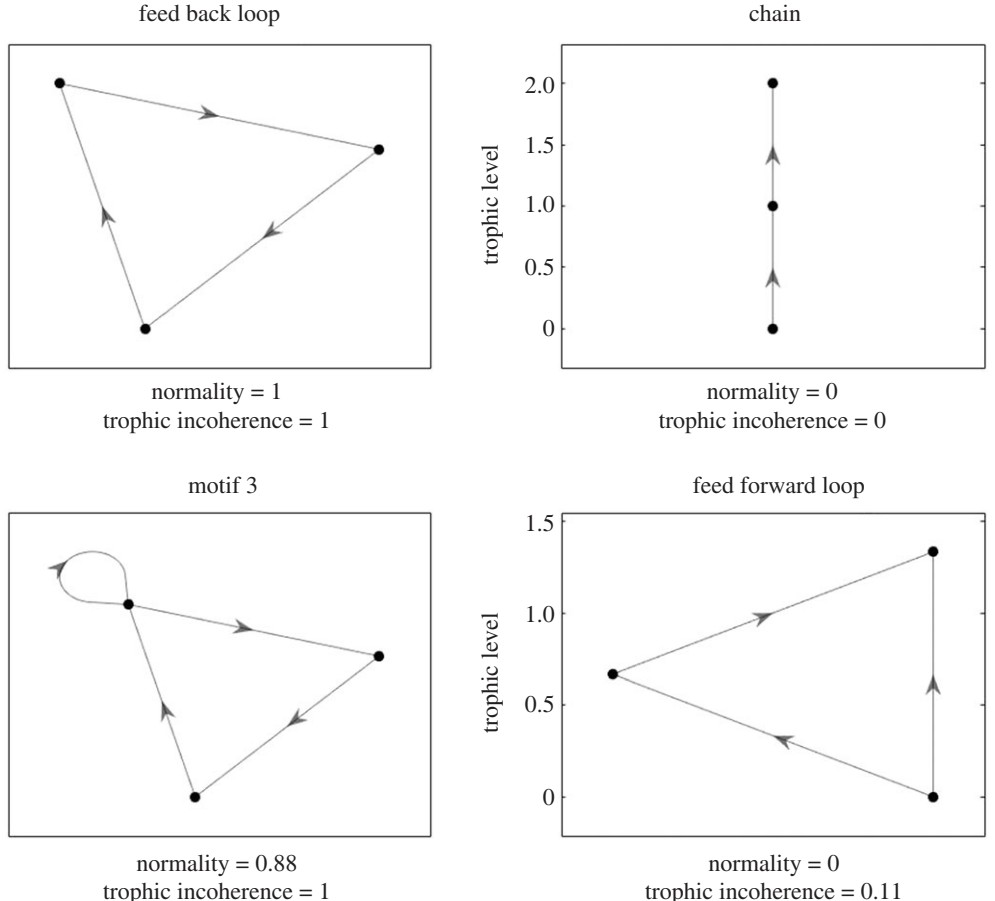

**Figure 11.** Four simple motifs illustrating relationship between incoherence and normality: for the unweighted case of an adjacency matrix $A$, normality ($\nu = 1$) implies the imbalance vector $v = 0$, thus $F_0 = 1$. This is illustrated by the feed-back loop (top left). However, maximal incoherence is not equivalent to normality—motif 3 (bottom left) demonstrates one can have $F_0 = 1$ without $\nu = 1$ (here motif 3 (6.2) is non-normal ($\nu = 0.88$)). If $W$ is maximally coherent ($F_0 = 0$) then all its eigenvalues are 0, so $\nu = 0$ and it is maximally non-normal. This is illustrated by the chain (top right). However, one can have $\nu = 0$ without $F_0 = 0$. This is demonstrated by the feed-forward motif (bottom right), which has $\nu = 0$ but $F_0 = 0.11$.

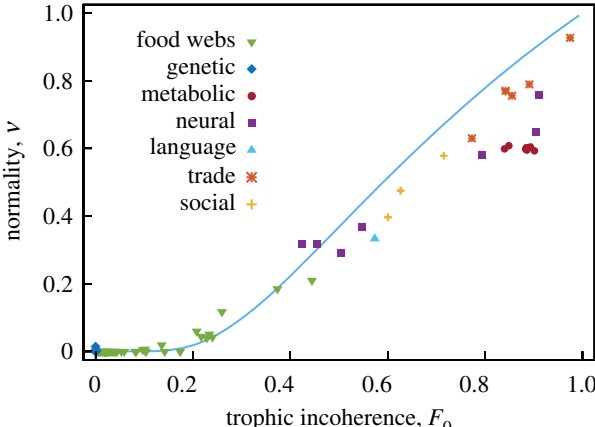

**Figure 12.** Normality $\nu$ against trophic incoherence $F_0$ for some networks. The curve corresponds to the coherence-ensemble expectation $\overline{\nu} = \exp(1 - 1/F_0)$.

network will grow or decay. In vector–matrix form, the solution after time $t \in \mathbb{Z}_+$ is

$$x(t) = \frac{x(0)\,W^t}{r^t}. \tag{6.6}$$

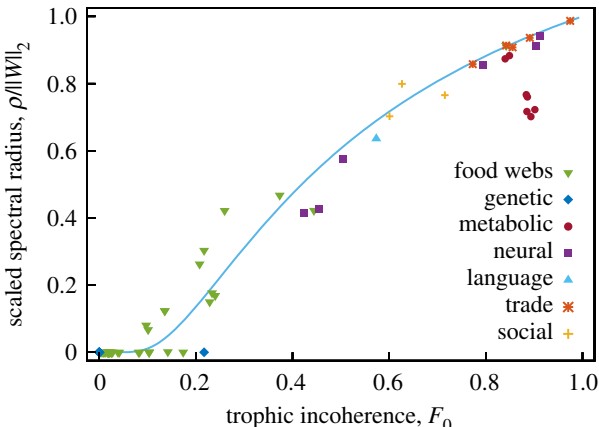

**Figure 13.** Scaled spectral radius $\rho/\|W\|_2$ against trophic incoherence $F_0$ for some networks. The curve corresponds to the coherence-ensemble prediction of $\rho_s = \exp((1 - 1/F_0)/2)$.

The answer (see appendix A.8) is that if $\rho < r$ then $\|x(t)\|_1 \to 0$ as $t \to \infty$, whereas if $\rho > r$ and *condition* K holds ($x_n > 0$ for some node $n$ in or leading to a 'key' communicating class), then $\|x(t)\|_1 \to \infty$, where the *spectral radius* $\rho$ of $W$ is the largest absolute value of the eigenvalues of $W$. Actually, because $W$ has all entries non-negative, it has a real positive eigenvalue of maximum modulus, so that is $\rho$. Indeed, under condition $K$,

$$t^{-1} \log \|x(t)\|_1 \to \log\left(\frac{\rho}{r}\right) \text{as } t \to \infty. \tag{6.7}$$

We have already mentioned that a maximally coherent network has all its eigenvalues 0, so $F_0 = 0$ implies $\rho = 0$. This suggests that $\rho$, scaled by a suitable measure of the strength of $W$, might correlate positively with $F_0$. The strength of $W$ can be measured by any norm, for example the 2-norm $\|W\|_2$. This can be defined in various ways, of which perhaps the simplest is that $\|W\|_2^2$ is the largest eigenvalue of $W^T W$ (which is necessarily real and non-negative and is equal to that for $WW^T$). For any operator-norm, $\rho \le \|W\|$. Thus $\rho/\|W\|$ is contained in [0, 1], like $F_0$. An advantage of the particular choice of the 2-norm is that $\rho = \|W\|_2$ if $W$ is normal. So we define the *scaled spectral radius*

$$\rho_s = \frac{\rho}{\|W\|_2}. \tag{6.8}$$

Then we deduce from the subsection on normality various cases with simultaneously $F_0 = 1$ and $\rho_s = 1$.

Thus we look at how $F_0$ correlates with the scaled spectral radius $\rho_s$ in figure 13. In appendix A.11, we give heuristic arguments in favour of a relation

$$\rho_s \approx \exp\left(\frac{1}{2}\left(1 - \frac{1}{F_0}\right)\right). \tag{6.9}$$

We can also consider a simple dynamical model for contagion in continuous time,

$$\dot{x}_n = \sum_m x_m w_{mn} - r x_n, \tag{6.10}$$

with $r$ a recovery rate (without immunity). The solution can be written in vector-matrix form as

$$x(t) = x(0)\, e^{(W - rI)t}. \tag{6.11}$$

Again one can ask whether the total infection $\|x(t)\|_1$ grows or decays. This is now a question of the maximal real part of the eigenvalues of $W$, but because $W$ is non-negative, the maximal real part of eigenvalues is actually $\rho$. So the answer is growth for $\rho > r$, decay for $\rho < r$. So again it is interesting to link $\rho$ with $F_0$ and relation (6.9) will be useful.

Some other dynamics on networks are discussed in appendix A.9.

## 6.3. Cycles

By a *cycle* in a directed network, we mean a closed walk in it. A *walk* is any sequence $(e_j)_{j=1}^J$ of edges such that for $1 \leq j < J$ the head of $e_j$ is the tail of $e_{j+1}$. It is *closed* if the head of $e_J$ is the tail of $e_1$. In contrast to much of the graph-theory literature, we allow a cycle to have repeated edges and repeated nodes, but we prefer to use the shorter and more familiar word 'cycle' than 'closed walk'. In particular, we allow a cycle to be a periodic repetition of a shorter cycle. The *weight* $w_\gamma$ of a cycle $\gamma$ is the product of the weights along its edges.

A maximally coherent network ($F_0 = 0$) has no cycles, because it has height difference $+1$ for every edge, whereas along a cycle the net change in height has to be zero. There are acyclic graphs with $F_0 > 0$, however, for example the feed-forward motif (6.4).

A maximally incoherent network ($F_0 = 1$) must have cycles. This is because it is balanced and so some of the flow that leaves a node must eventually come back to it (see appendix A.12). In fact, we deduce that every edge is in at least one cycle.

So these results suggest some relation between trophic incoherence $F_0$ and a quantifier of cyclicity.

The total weight of cycles of length $p$ is given by the trace of the $p^{th}$ power of $W$: $\operatorname{tr} W^p$, because $(W^p)_{mn} = \sum_j w_{n_0 n_1} \dots w_{n_{p-1} n_p}$ and the trace of a matrix is the sum of its diagonal entries. Note that this counts each cyclic permutation of a cycle as a different cycle. One might expect it to behave asymptotically exponentially as $p \to \infty$, but for example if $k$ points in a circle are each connected to just their clockwise neighbour by an edge of weight $x$, then $\operatorname{tr} W^p = kx^p$ when $p$ is a multiple of $k$, $0$ otherwise. The tidy way to study the sequence $\operatorname{tr} W^p$ is to form the *zeta function*

$$\zeta(z) = \exp \sum_{p=1}^{\infty} \frac{z^p}{p} \operatorname{tr} W^p, \tag{6.12}$$

for complex $z$ close enough to 0 (some authors define $\zeta(z)$ to be the reciprocal of this). Then a notion of the cyclicity of $W$ is the reciprocal of the radius of convergence of the power series. This is just $\limsup_{p\to\infty} (\operatorname{tr} W^p)^{1/p}$. Using $\log \det = \operatorname{tr} \log$, the zeta function can equivalently be written as $\det(I - zW)^{-1}$. The reciprocal of its radius of convergence is the spectral radius $\rho$. So actually, the appropriate measure of cyclicity is $\rho$ relative to some measure of the size of $W$. We take again $\|W\|_2$ for the latter. Thus, cyclicity $\rho/\|W\|_2 = \rho_s$ is related to $F_0$ exactly as is the stability of our simple contagion processes. In particular it is 1 for any normal network.

In appendix A.13, we relate $\zeta$ to the *prime* cycles, those which are not repetitions of a shorter cycle, and furthermore to the *elementary* cycles, those which do not repeat a node.

# 7. Extension to arbitrary target height differences

So far, we have taken all the target height differences equal, but there are contexts in which this might not be appropriate. Instead of trying to fit the height differences along each edge to 1, it might be preferable to fit them to more general target height differences $\tau_{mn}$. For example, if two nodes $m$, $n$ are subunits of a single company with $m$ supplying $n$ it might be reasonable to assign a value $\tau_{mn}$ less than 1. Or if there is a feed-forward motif that one does not want to contribute to circularity, then one could assign target height difference $1/2$ to the edges for the indirect route and 1 to the direct edge.

The extension of our method to this setting is straightforward. Minimize

$$F(h) = \frac{\sum_{mn} w_{mn}(h_n - h_m - \tau_{mn})^2}{\sum_{mn} w_{mn}\tau_{mn}^2},$$

over $h$. This is equivalent to solving $\Lambda h = v$ with now $v_n = \sum_m (w_{mn}\tau_{mn} - w_{nm}\tau_{nm})$. If we set $F_0$ to be the minimizing value of $F$ and $z_{mn} = (h_n - h_m)/\tau_{mn}$ and weight the $z_{mn}$ by $w_{mn}\tau_{mn}^2$, then we obtain all the same properties as before: $0 \leq F_0 \leq 1$, $\bar{z} = 1 - F_0$, etc.

One context in which it is natural to use the data to set the target height differences is quantitative pairwise comparison, for example, use of goal differences in a football league to infer relative strengths of teams. Then it is natural to make each game carry equal weight and the outcome of our extended method is trophic levels representing the relative strengths of the teams [47].

# 8. Discussion

In many domains of science, one is faced with a directed network and wishes to determine (i) to what extent the edges line up in an overall direction, and (ii) the relative position of individual nodes within any directional flow on the network.

The old concept of 'trophic level' from ecology, and its more recent analogue 'upstreamness' in economics, provided an answer to the question of the relative positions of individual nodes; however, these previous notions required (respectively) basal or top nodes; they give too much emphasis to basal/top nodes if there is more than one; and they do not give a stable way to determine levels and incoherence for a piece of a network.

The notion of 'trophic coherence' introduced in [7], based on the old notion of trophic levels, provides a way to quantify the extent to which edges align in an overall direction and was shown to be connected with network properties such as cycles and spectral radius, but it lacks a natural notion of maximal incoherence.

In this paper, we have introduced improved notions of trophic level and trophic coherence, which do not require basal or top nodes; are as easy to compute as the old notions; and are connected in the same way with network properties such as normality, cycles and spectral radius. Furthermore, they remove the problem of bias from basal nodes, make incoherence have a natural range from perfect coherence to maximal incoherence, and make it possible to compute them locally in a network without having to compute them for the whole network.

We expect this to be a valuable tool in domains from ecology, gene expression, neuroscience and biochemistry to economics, finance, social science and humanities.

Data accessibility. The data used to produce figure 1 were downloaded from [48] and can be accessed from: https://datadryad.org/stash/dataset/doi:10.5061/dryad.1mv20r6. The data used to produce figures 2 and 3 are from the OECD Input-Output Tables described and available here: http://www.oecd.org/sti/ind/input-outputtables.htm, from the OECD website. The data for figure 4 were downloaded from [20] here: https://www.ncbi.nlm.nih.gov/pmc/articles/PMC2736650/. The data for figure 5 published in [25] were downloaded from http://language.media.mit.edu. The supply network datasets presented in figures 7 and 8 and used for the analysis presented in figure 10 are based on supply-chain relationships compiled from Bloomberg L.P. supply chain function. Bloomberg's database compiles information from a wide variety of sources to provide a view of global supply chains at the firm level. More information on these data can be obtained from Bloomberg L.P. or [49]. To construct our networks of supplier–buyer relationships, starting from a focal firm of interest we then followed links identified by the Bloomberg database. These datasets could with Bloomberg's permission be made available on request, or re-compiled from Bloomberg. The data used for figures 12 and 13 can be downloaded from https://www.samuel-johnson.org/data, along with a list with references to the original sources. All code used to make empirical and computational analysis of public data and data-files is available for download at this Github repository, where we also provide a Matlab toolbox for the easy implementation of the methods we have introduced and related analysis.

Authors' contributions. R.S.M. obtained the grant for the project, came up with the improved notions of trophic level and incoherence and proved most of the results about them. S.J. did tests on a variety of networks, the comparisons of the new notion of trophic incoherence with other quantifiers of network structure, the ensemble theory for them, and some of the other proofs. B.S. built the Matlab toolbox, and did many of the empirical tests reported here, including all those on supply networks and input-output networks. Each of us wrote parts of the text and contributed to its finalization.

Competing interests. R.S.M. is an Associate Editor of RSOS but played no role in its assessment.

Funding. The support of the Economic and Social Research Council (UK), through grant no. ES/R00787X/1, is gratefully acknowledged. This was awarded via a call from the Instability hub of the Rebuilding Macroeconomics programme of the National Institute for Economic and Social Research (NIESR). S.J. and R.S.M. acknowledge support of the Alan Turing Institute under EPSRC grant no. EP/N510129/1 and Fellowship grant no. TU/B/000101.

Acknowledgements. We are grateful to other members of our NIESR project team for their comments, especially Nicholas Beale for asking for a quantification of coherence that does not depend on having basal nodes and has a clear maximum and minimum, and to other members of the Instability hub for their comments and interest. We are also grateful to Giannis Moutsinas and Choudhry Shuaib for sharing their approach to the subject, to Mark Pollicott for useful discussion about the zeta function, and to reviewers for insightful comments and pointers to the wider literature.

# Appendix A

## A.1. Solutions of $\Lambda h = v$

The graph Laplacian $\Lambda$ is not invertible: for any constant vector $h$, $\Lambda h = 0$. Indeed for any $h$ that is constant on connected components of the network, $\Lambda h = 0$, and the kernel of $\Lambda$ is precisely this set of $h$. Similarly,

for any $h$, the components of $\Lambda h$ on each connected component of the network add up to zero, and this property characterizes the range of $\Lambda$. Now the imbalance vector $v$ has the special property that the sum of its components over any connected component of the network is zero. Thus it follows that $\Lambda h = v$ always has a solution $h$, and the general solution is given by adding any vector that is constant on each connected component.

## A.2. Range for trophic incoherence $F_0$

Here we prove that $0 \leq F_0 \leq 1$ with $F_0 = 0$ iff all height differences $z_{mn} = 1$ and $F_0 = 1$ iff all height differences are 0.

First, we explain that the trophic heights $h$ solving $\Lambda h = v$ correspond to the minima of the trophic confusion function

$$F(h) = \frac{\sum_{mn} w_{mn}(h_n - h_m - 1)^2}{\sum_{mn} w_{mn}}, \tag{A 1}$$

over all possible assignments of heights $h_n$, $n \in N$. This is because the second derivative of $F$ is positive semi-definite, so all critical points are minima, and by differentiating with respect to each $h_n$, the equation for critical points is $\Lambda h = v$. Furthermore, the minimum value of this expression is $F_0$.

Since $F(h) \geq 0$ for all $h$, we see that $F_0 \geq 0$. Furthermore, $F_0 = 0$ iff all height differences are 1. Next, putting all heights equal, say to 0, denoted by $\mathbf{0}$, gives $F(\mathbf{0}) = 1$, so $F_0 \leq 1$. Now if $F_0 = 1$ at some $h$ then because $F(\mathbf{0}) = 1$ and the second derivative of $F$ is positive semi-definite with null space given by constants on each connected component, then $h - \mathbf{0}$ must be in this nullspace, i.e. $h$ is constant on each connected component. Thus, all height differences along edges are zero.

## A.3. Mean height difference

The mean height difference

$$\bar{z} = \frac{\sum_{mn} w_{mn}(h_n - h_m)}{\sum_{mn} w_{mn}}, \tag{A 2}$$

is $1 - F_0$. To prove this, write the trophic confusion function as

$$F(h) = \sigma^2 + \bar{z}^2 - 2\bar{z} + 1, \tag{A 3}$$

with

$$\sigma^2 = \frac{\sum_{mn} w_{mn}(h_n - h_m - \bar{z})^2}{\sum_{mn} w_{mn}}. \tag{A 4}$$

If $h$ minimizes $F$ then $F(\alpha h)$ must be minimized over $\alpha \in \mathbb{R}$ at $\alpha = 1$. But

$$F(\alpha h) = \alpha^2(\sigma^2 + \bar{z}^2) - 2\alpha\bar{z} + 1, \tag{A 5}$$

which has unique minimum at $\alpha = \bar{z}/(\sigma^2 + \bar{z}^2)$ (unless $\sigma = \bar{z} = 0$). Thus $\bar{z} = \sigma^2 + \bar{z}^2$. But $\sigma^2 + \bar{z}^2 - 2\bar{z} + 1 = F_0$. So $\bar{z} = F_0 + 2\bar{z} - 1$, hence $\bar{z} = 1 - F_0$. If $\sigma = \bar{z} = 0$, we see that $F_0 = 1$ and hence $\bar{z} = 1 - F_0$ is satisfied in that case too.

## A.4. Electrical interpretation

Our notion of trophic levels can be given an electrical interpretation. The edge weights are conductivities of bidirectional connectors between nodes. Current $v_n$ is injected into (or extracted from, according to sign) each node $n$. The resulting voltages (modulo an arbitrary overall shift) are the trophic heights $h_n$. One could imagine the currents $v_n$ as being generated by making a copy of all the incoming and outgoing edges of node $n$ and imposing a voltage difference of $+1$ on all its input nodes and $-1$ on all its output nodes, relative to $n$.

## A.5. Robustness of trophic levels to truncation of the network

We recall that we choose a connected subset called zone 1 and fix the height of one of its nodes (or the weighted average of its nodes) to be 0. We choose a buffer zone 2 so that there are no direct connections between zone 1 and the outside, called zone 3, and so that the union of zones 1 and 2 is connected.

Then the equation $\Lambda h = v$ for the heights can be broken into the block form

$$\Lambda_{11}h_1 + \Lambda_{12}h_2 = v_1, \tag{A 6}$$

$$\Lambda_{21}h_1 + \Lambda_{22}h_2 + \Lambda_{23}h_3 = v_2 \tag{A 7}$$

and

$$\Lambda_{32}h_2 + \Lambda_{33}h_3 = v_3. \tag{A 8}$$

Changes to the outside zone 3 can affect $v_2$ and the diagonal part of $\Lambda_{22}$. Let us suppose that the total weights of connections in each direction between zone 3 and each node of 2 are given. Thus, $v_2$ and $\Lambda_{22}$ are fixed. Let $\bar{h}$ be the solution for the reference case where all of zone 3 is amalgamated to a single node. By the connectedness assumption, $\bar{h}$ exists and is unique up to an overall shift. Let $\tilde{h} = h - \bar{h}$ with $h$ the solution for the true zone 3, subtracting the single number $\bar{h}_3$ from each element of $h_3$. Then

$$\Lambda_{11}\tilde{h}_1 + \Lambda_{12}\tilde{h}_2 = 0 \tag{A 9}$$

and

$$\Lambda_{21}\tilde{h}_1 + \Lambda_{22}\tilde{h}_2 + \Lambda_{23}\tilde{h}_3 = 0. \tag{A 10}$$

By the connectedness of zone 1, $\Lambda_{11}$ is invertible modulo overall shifts, on the subspace such that the sum of the components is zero. We have taken care of overall shifts by fixing a node of zone 1 to be at height 0. The sum of the components of $\Lambda_{12}\tilde{h}_2$ is automatically zero, because taking the sum of (A 9) over components in zone 1, $\Lambda_{11}\tilde{h}_1$ gives 0. So

$$\tilde{h}_1 = -\Lambda_{11}^{-1}\Lambda_{12}\tilde{h}_2. \tag{A 11}$$

Similarly, by connectedness of the union of zones 1 and 2, and substituting the above,

$$\tilde{h}_2 = -(\Lambda_{22} - \Lambda_{21}\Lambda_{11}^{-1}\Lambda_{12})^{-1}\Lambda_{23}\tilde{h}_3. \tag{A 12}$$

Thus the desired answer is

$$\tilde{h}_1 = \Lambda_{11}^{-1}\Lambda_{12}(\Lambda_{22} - \Lambda_{21}\Lambda_{11}^{-1}\Lambda_{12})^{-1}\Lambda_{23}\tilde{h}_3. \tag{A 13}$$

Thus by taking norms throughout (for example, the weighted sum $\|h\| = \sum_n u_n|h_n|$ and the corresponding operator norm), we obtain a bound on the changes to the levels on zone 1 in terms of a bound on the changes to the levels on the part of zone 3 connecting directly to zone 2:

$$\|\tilde{h}\| \leq \|\Lambda_{11}^{-1}\|\|\Lambda_{12}\|\|(\Lambda_{22} - \Lambda_{21}\Lambda_{11}^{-1}\Lambda_{12})^{-1}\|\|\Lambda_{23}\tilde{h}_3\|. \tag{A 14}$$

The latter is unknown in general, but the formula gives some idea of how much the levels change on zone 1 on incorporating more detail about zone 3. In particular, if zone 1 is well connected in the sense that $\|\Lambda_{11}^{-1}\|$ is not large, and zones 1 and 2 are well connected in the sense that $\|(\Lambda_{22} - \Lambda_{21}\Lambda_{11}^{-1}\Lambda_{12})^{-1}\|$ is not large then $\tilde{h}_1$ is not very sensitive to changes $\tilde{h}_3$ to the levels in zone 3.

An alternative to fixing the height of a node in zone 1 is to consider height vectors as equivalent if they differ by an overall shift and use a norm that pays attention only to height differences, e.g. $\|h\| = \sum_{mn} w_{mn}|h_n - h_m|$.

## A.6. Balanced iff maximally incoherent

If $v = 0$ then $\Lambda h = 0$ so $h$ is constant on connected components, so $F_0 = 1$. Conversely, if $F_0 = 1$ then $h$ is constant on connected components, so $v = \Lambda h = 0$.

Note that it follows that maximally incoherent networks have no basal nodes (more precisely, any basal node is connected to no other nodes).

## A.7. Maximal coherence implies normality zero

If $W$ is maximally coherent then the level difference for each edge is $+1$ so, arranging the nodes in order of height, the matrix $W$ is upper triangular with zero diagonal. It follows that all its eigenvalues are 0. Hence $v = 0$.

## A.8. Stability of contagion processes

For $x(t) = x(0)W^t/r^t$, we have

$$\|x(t)\|_1 \leq \frac{\|x(0)\|_1 \|W^t\|_1}{r^t}, \tag{A 15}$$

using the induced operator-norm on $W$, so

$$t^{-1} \log \|x(t)\|_1 \leq t^{-1} \log \|x(0)\|_1 + t^{-1} \log \|W^t\|_1 - \log r. \tag{A 16}$$

But for any operator-norm $t^{-1}\log\|W^t\| \to \log \rho$ as $t \to \infty$ [50]. So if $\rho < r$ then $\|x(t)\|_1 \to 0$ as $t \to \infty$.

In the other direction, we need theory for non-negative matrices $W$, e.g. [51]. A node in a directed graph is *recurrent* if there is a cycle through it. Two recurrent nodes communicate if there is a cycle through both. The set of recurrent nodes can be decomposed into *communicating classes*, subsets in which each pair of nodes communicate and between which no pair of nodes communicate. The eigenvalues of $W$ consist of the eigenvalues of its restrictions $W_c$ to each communicating class $c$ and an eigenvalue 0 for each non-recurrent node. The period $P$ of a communicating class $c$ is the highest common factor of the lengths of all cycles in it. The communicating class $c$ can be decomposed into $P$ *cyclic classes*, whose nodes can only be reached from each other in a multiple of $P$ steps. They can be labelled $c_0, \ldots c_{P-1}$ so that one can get from $c_j$ to $c_k$ only in a number of steps congruent to $k-j$ modulo $P$.

On each cyclic class $c_j$, the restriction of $W^P$ is irreducible and aperiodic. So by Perron–Frobenius theory [51] it has a simple positive eigenvalue $\lambda_1$ with positive eigenvector, and the remaining eigenvalues satisfy $|\lambda_k| < \lambda_1$. Throughout this item, we consider left eigenvectors because we are interested in the action of $W$ on row-vectors $x$. The eigenvalues of $W_{c_j}^P$ on the cyclic classes of $c$ are related as follows. If $xW_c^P = \lambda x$ with $x$ supported on $c_0$ and non-zero, then $xW_c$ is supported on $c_1$ and $(xW_c)W_c^P = xW_c^P W_c = \lambda xW_c$, so either $xW_c$ is an eigenvector for $W_c^P$ on $c_1$ with the same eigenvalue or it is zero. If $xW_c = 0$ then $\lambda x = xW_c W_c^{P-1} = 0$ so $\lambda = 0$. Thus $W_{c_j}^P$ have the same eigenvalues apart from possible 0s. If the cyclic classes have different sizes, eigenvalues 0 must occur for all but the smallest ones.

From the non-zero eigenvalues $\lambda$ of $W_{c_j}^P$, we deduce that the non-zero eigenvalues of $W_c$ are the (complex) $P^{th}$ roots of $\lambda$ as follows. Take an eigenvector $x$ on $c_0$ for $\lambda \neq 0$. Let $\zeta$ be any $P^{th}$ root of $\lambda$. Then $[\zeta^P x, \zeta^{P-1}xW_c, \ldots \zeta xW_c^{P-1}]$ is an eigenvector of $W_c$ with eigenvalue $\zeta$, where the components in the vector are grouped according to the cyclic classes $c_0, \ldots c_{P-1}$. So the eigenvalues of $W_c$ are the $P^{th}$ roots of the non-zero eigenvalues of $W_{c_0}^P$, augmented by 0s. The eigenvectors of $W_c$ can be extended to eigenvectors of $W$ on the whole network with the same eigenvalue.

If $x(0) \geq 0$ is positive on some node of a cyclic class $c_j$ of a communicating class $c$ then by Perron–Frobenius theory,

$$\lambda_1^{-n} x(0) W_{c_j}^{nP} \to C\hat{x} \text{ as } n \to \infty, \tag{A 17}$$

for some $C > 0$, where $\hat{x}$ is the Perron–Frobenius eigenvector on $c_j$ and $\lambda_1$ its eigenvector. Furthermore $\lambda_1^{-n} x(0) W_c^{nP+s} \to C\hat{x}W_c^s$. Including the rest of the edges,

$$\lim_{n\to\infty} t^{-1} \log \|x(t)\|_1 \geq P^{-1} \log \lambda_1. \tag{A 18}$$

We say a communicating class is 'key' if its $\lambda_1 = \rho^P$. There is always at least one such. Thus if $x(0)$ is positive on some node of a key communicating class, then, combining with (A 16) and (A 18),

$$t^{-1} \log \|x(t)\|_1 \to \log\left(\frac{\rho}{r}\right) \text{as } t \to \infty. \tag{A 19}$$

Similarly, if $x(0)$ is positive on a node leading to a key communicating class, then in finitely many steps $x(t)$ is positive on some node of that class and hence the same result follows.

## A.9. Other dynamics

The context in which trophic coherence was first proposed [7] is that of Lotka–Volterra dynamics for populations of species in an ecosystem. This is somewhat difficult to treat because if $W$ quantifies how much one species eats of another this does not give a complete specification of the population dynamics. But as in [7], one can propose

$$\dot{x}_n = x_n \left( r_n - \sum_m w_{nm} x_m + \eta \sum_m x_m w_{mn} - \kappa_n x_n \right), \tag{A 20}$$

where $r_n$ is a natural birth or death rate (according to sign) for species $n$, the negative sum accounts for species $n$ being eaten, the positive sum accounts for the enhancement of population of species $n$ from what it eats, with an efficiency factor $\eta$, and the final term accounts for effects of intraspecies competition not included in cannibalism ($w_{nn}$). Write it in the form

$$\frac{\mathrm{d}}{\mathrm{d}t} \log x = r - Bx, \tag{A 21}$$

where $\log x$ stands for the vector with components $\log x_n$.

One first question is whether this has any positive equilibria. The equilibria are given by choosing any subset of species to be extinct and the rest to satisfy $Bx = r$ where the rows and columns corresponding to extinct species have been deleted. To be physical, the remaining components of $x$ must all be positive.

Given a positive equilibrium $x$, possibly of a subsystem given by deleting extinct species, a second question is whether it is stable. The linearized equations for deviations $\xi$ from an equilibrium are

$$\dot{\xi}_n = -x_n \sum_m B_{mn} \xi_n. \tag{A 22}$$

So even if we know $B$, the linearized equations are not completely determined because we need to know the equilibrium $x$.

Similarly, economic dynamics can be proposed on supply networks [52] and the question arises whether there is a relation between stability and trophic coherence.

## A.10. Ensemble relation of normality to incoherence

It is possible to relate trophic coherence with various other topological features by considering ensembles of random graphs [10]. The 'coherence ensemble' is the set of all unweighted, directed networks with given in- and out-degree sequences and given trophic coherence. For example, using the standard definition of trophic incoherence $q$, the expected value of the spectral radius $\rho$ in the coherence ensemble is

$$\overline{\rho} = \mathrm{e}^{\tau}, \tag{A 23}$$

where

$$\tau = \ln \alpha + \frac{1}{2\hat{q}^2} - \frac{1}{2q^2}, \tag{A 24}$$

(and we use a bar to represent coherence-ensemble expectation). Here, $\hat{q}$ is the expected trophic incoherence in the 'basal ensemble', and $\alpha = \langle w^{\mathrm{in}} w^{\mathrm{out}} \rangle / \langle w \rangle$ is the branching factor, but for current purposes we need not discuss these magnitudes in detail. In previous work, the trophic coherence was measured with the incoherence parameter $q$, which corresponds to the standard deviation over trophic differences when the average trophic difference is 1. Using the new definition of levels we are proposing here, the equivalent of this magnitude is

$$\eta = \sqrt{\frac{F_0}{1 - F_0}}, \tag{A 25}$$

as given in the main text.

We note that the ratio between the expected spectral radius for a given coherence, $\overline{\rho}$, and the value corresponding to a maximally incoherent network,

$$\overline{\rho}_{\max} = \lim_{\eta \to \infty} \mathrm{e}^{\tau}, \tag{A 26}$$

depends only on trophic coherence:

$$\frac{\overline{\rho}}{\overline{\rho}_{\max}} = \exp\left( -\frac{1}{2\eta^2} \right). \tag{A 27}$$

In the main text we measure normality with

$$\nu = \frac{\sum_j |\lambda_j|^2}{\|W\|_F^2}. \tag{A 28}$$

A normal network (if unweighted) is, as described in the main text, a balanced network, which is maximally incoherent ($F_0 = 1$). In this case, we have $\nu = 1$. On the other hand, the greatest deviation from normality is achieved when $|\lambda_j| = 0$ for all $j$, which is the case of maximally coherent networks ($F_0 = 0$). For networks in the coherence ensemble with $0 \leq F_0 \leq 1$, we postulate that

$$\frac{\sum_i |\lambda_i|^2}{\|W\|_F^2} \simeq \frac{\rho^2}{\rho_{max}^2}, \tag{A 29}$$

which amounts to assuming that the distribution of eigenvalues of $W$ within the spectral radius $\rho$ does not depend on trophic coherence. This argument uses (i) $\|W\|_F^2 = \mathrm{tr}\ W^T W$, which in turn is the sum of the eigenvalues of $W^T W$, (ii) for $W$ normal the eigenvalues of $W^T W$ are precisely the squares of the absolute values of the eigenvalues of $W$, and (iii) normality is almost equivalent to maximal incoherence, as already discussed. Combining this expression with equations (A 25), (A 27) and (A 28), we have an approximate expression for the expected normality in the coherence ensemble,

$$\overline{\nu} \simeq \exp\left(1 - \frac{1}{F_0}\right). \tag{A 30}$$

Figure 12 shows $\nu$ against $F_0$ for our set of empirical networks, alongside equation (A 30). The empirical values fall fairly close to the ensemble expectations, with high coherence corresponding to a maximal non-normality, and incoherence being associated with greater normality. In many cases, the real networks are somewhat less normal than the ensemble prediction. This might be because these are relatively small networks in which statistical fluctuations play a large role, and at intermediate values of trophic coherence there are more ways of being non-normal than normal.

## A.11. Ensemble relation with scaled spectral radius

Using the results for the coherence ensemble again, in particular equations (A 27) and (A 25), we obtain that

$$\rho_s = \frac{\rho}{\|W\|_2} \simeq \frac{\overline{\rho}}{\overline{\rho}_{max}} = \exp\left(\frac{1}{2}\left(1 - \frac{1}{F_0}\right)\right). \tag{A 31}$$

Here, no assumption on the distribution of the eigenvalues of $W$ is required, simply the fact that $\rho = \|W\|_2$ for maximally incoherent networks.

The fit in figure 13 is again reasonable.

## A.12. Maximal incoherence implies cycles

A maximally incoherent network is balanced. Make a volume-preserving dynamical system in continuous time by converting each edge $mn$ to a tube of volume $V_{mn} > 0$ of incompressible fluid with flow rate $w_{mn}$ from $m$ to $n$, splitting the resulting flow into $n$ in any way between the out-edges of $n$ consistent with their weights. If none of the fluid originally in tube $mn$ comes back to that tube then after time $T$, tube $mn$ has ejected a volume $w_{mn}T$ of fluid that has to fit in the volume $\sum V_{jk}$ of the other tubes. But that is finite, so for $T$ large enough we get a contradiction. Hence there is a cycle through $mn$. So each edge of a maximally incoherent network is on a cycle.

One could allow the nodes to have volume too. The same argument works for countably infinite networks, by choosing the volumes to have a finite sum.

## A.13. Zeta function

The zeta function of the main text is a weighted version of the Bowen–Lanford zeta function (described in section 3.1 of [53]). It can be related to the *prime* cycles, those which are not repetitions of a shorter cycle. We consider two prime cycles to be the same if they differ only by a cyclic permutation. We denote by $\mathcal{P}$

the set of prime cycles. The formula is

$$\zeta(z) = \prod_{\gamma \in \mathcal{P}} (1 - z^{|\gamma|} w_\gamma)^{-1},$$ (A 32)

where $|\gamma|$ is the length of $\gamma$ and $w_\gamma$ its weight.

Here is a proof of the identity (cf. [53]).

$$\log \prod_\gamma (1 - z^{|\gamma|} w_\gamma)^{-1} = -\sum_\gamma \log (1 - z^{|\gamma|} w_\gamma)$$

$$= \sum_\gamma \sum_{k \geq 1} \frac{(z^{|\gamma|} w_\gamma)^k}{k} = \sum_{k \geq 1} \sum_\gamma |\gamma| \frac{z^{k|\gamma|}}{k|\gamma|} w_\gamma^k$$ (A 33)

$$= \sum_{p \geq 1} \frac{z^p}{p} \operatorname{tr} W^p,$$

because every cycle is a repetition of some prime cycle $\gamma$, say $k$ times, its weight is $w_\gamma^k$ and there are $|\gamma|$ cyclic permutations of it. The last expression is $\log \zeta(z)$, concluding the proof.

Equation (A 32) can be reduced to one in terms of 'elementary cycles', those which do not repeat a node before closing. They are prime and for a finite network there are only finitely many of them. The formula is

$$\frac{1}{\zeta(z)} = 1 + \sum_C \prod_{\gamma \in C} (-z^{|\gamma|} w_\gamma),$$ (A 34)

where the sum is over non-empty collections $C$ of disjoint elementary cycles. This is a clean case of Cvitanovic's cycle expansions [54], and appeared already in eqn (18.13) of [55]. We give a proof, however, because it seems to us that [55] left out the case of self-edges.

To prove (A 34), use $1/\zeta(z) = \det(I - zW)$ and the formula

$$\det M = \sum_{\pi \in S_n} \epsilon_\pi M_{1\pi_1} \ldots M_{n\pi_n},$$ (A 35)

for an $n \times n$ matrix $M$, where $S_n$ is the group of permutations of $\{1, \ldots n\}$ and $\epsilon_\pi$ is the sign of the permutation $\pi$ (+1 if $\pi$ can be written as an even number of transpositions, −1 for an odd number). For $M = I - zW$, the only permutations for which the product in (A 35) is non-zero are those which can be written as a product of disjoint cyclic permutations corresponding to elementary cycles of period at least 2 in the network and the identity permutation on the remaining nodes. The contribution of a collection $C_2$ (possibly empty) of disjoint elementary cycles of period at least 2 is

$$\prod_{m \in C'} (1 - zw_{mm}) \prod_{\gamma \in C_2} (-z^{|\gamma|} w_\gamma),$$ (A 36)

where $C'$ is the set of nodes not in $C_2$. If there are no self-edges then $w_{mm} = 0$ for all $m$ and there are no cycles of period 1, so adding in the case of the empty collection, we obtain (A 34) when there are no self-edges.

If there are some self-edges then expand out (A 36) to

$$\sum_{C_{2+}} \prod_{\gamma' \in C_{2+}} (-z^{|\gamma'|} w_{\gamma'}),$$ (A 37)

where the sum is over collections $C_{2+}$ of disjoint elementary cycles formed by adding any 1-cycles to $C_2$, including the case of adding no 1-cycles. Lastly, the contribution of the identity permutation is

$$\prod_m (1 - zw_{mm}) = 1 + \sum_{C_1} \prod_{\gamma \in C_1} (-zw_\gamma),$$ (A 38)

where the sum is over non-empty collections of disjoint 1-cycles. Adding together (A 37) and (A 38), we obtain the result (A 34) for the general case.

## A.14. Relation to Helmholtz–Hodge decomposition

Iyetomi *et al.* [34] and Kichikawa *et al.* [35] obtain levels and a notion of circularity by a graph version of the Helmholtz–Hodge decomposition of a vector field into a gradient part and a conservative

(divergence-free) part. Here we show the equivalence of our method to theirs. For maximum generality, we do this in the context of arbitrary target height-differences (see §7).

They take flows $F_{mn}$ and conductivities $w_{mn}$ for each edge and decompose $F$ into $F^p + F^c$, with $F_{mn}^p = w_{mn}(\phi_m - \phi_n)$ and $F^c$ balanced, by minimizing

$$\sum_{mn} w_{mn}^{-1}(F_{mn} - w_{mn}(\phi_m - \phi_n))^2$$

over potentials $\phi$. We can write this in the form of our method, $\sum_{mn} w_{mn}(h_n - h_m - \tau_{mn})^2$, with $h_n = -\phi_n$ and

$$\tau_{mn} = \frac{F_{mn}}{w_{mn}}. \tag{A 39}$$

Thus there is a simple map (A 39) between (flows $F$, conductivities $w$) and (weights $w$, targets $\tau$). One just has to bear in mind that often we consider our weights $w$ to be flows, so the terminology can be confusing. They are the same in the main case treated in our paper (all targets = 1) and in [35] ($w = F$), but in general need distinguishing.

The approach of Kichikawa *et al.* [35] leads to alternative terminology, which can be advantageous. Their 'circularity' is our incoherence. A caveat, however, is that feed-forward motifs make a contribution to circularity. It would be interesting to devise a modified notion of true circularity, perhaps by automatic adjustment of the target height differences.

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
