## [Reviewer comments · Royal Society Open Science]

Review History

RSOS-201138.R0 (Original submission)

Review form: Reviewer 1

Is the manuscript scientifically sound in its present form?

Yes

Are the interpretations and conclusions justified by the results?

Yes

Is the language acceptable?

Yes

Do you have any ethical concerns with this paper?

No

Have you any concerns about statistical analyses in this paper?

No

Recommendation?

Major revision is needed (please make suggestions in comments)

Comments to the Author(s)

The manuscript is concerned with the introduction of a new measure of network incoherence which is based on a symmetrized graph Laplacian for weighted directed networks. The measure is tested on different real world data.

The manuscript is overall well written, but it is poorly organized and very wordy: it would benefit from reorganizing the material and incorporating in the text some of the results presented in the appendix. Moreover, many of the proofs could be carried out using formulas rather than words, and this simple switch would greatly improve readability. The comparison with other measures of incoherence should be carried out in a more thorough fashion, and it definitely deserve more space that it has been allocated in the manuscript. Disregarding directionality of edges is something that is usually best avoided, and I believe that the authors are not making a strong enough case for their decision to following this path in the manuscript.

I believe that the manuscript should be reconsidered for publication after a major revision.

There are a few points that I think are worth discussing and elaborating further on. Below I detail some of these.

p.2 eq (2.1): these quantities are extensios of the cocept of "strength", introduced by Barrat et al. in [PNAS (2004), 101(11),3747-3752]. I would recommend that the authors refer to this manuscript (and subsequent generalization to digraphs, if any) and that they use the term "in/out-strength" or similar, in order to emphasize the relationship with the index introduced in 2004.

p.2, eqs (2.2)-(2.3): what is the rationale for calling these "weight" and "imbalance"?

p.2. eq (2.4): this appears to be an extension to the weighted case of the a symmetrized graph Laplacian for digraphs (as can also be seen from equation (2.5)). This type of approach to the treatment of directed graphs is often criticised in the literature, as it completely changes the topology of the network (especially in the case of highly non-symmetric matrices W). Could the authors justify their approach and explain further why disregarding directionality of edges is the right thing to do in this context? In my opinion the justification is quite weak, as it stands. (This also links to the contents of page 9).

p.3 eq (2.5): please specify who the vector u is.

p.3 l.10: here h is characterized as being *the* solution to $\lambda h = v$ (please add specification of who v is). However, in l. 14 the authors state that $\lambda h = v$ does not have a unique solution. Please change l.10 to state that h is *a* solution and fully characterize $\text{span}\{h: \lambda h = v\}$.

p.3 l.14: Instead of considering the case of a disconnected network with several weakly connected components, it would be easier to just focus on the case of weakly connected networks; Then the matrix $W+W^T$ is the (weighted) adjacency matrix of a connected undirected graph and therefore the vector of all ones spans the null space of $W+W^T$. The case of disconnected networks follows from, e.g., chapter 6.13.3 in "Networks: An Introduction" by M. Newman. (Let me clarify that I understand that the authors are doing this already, I am only suggesting what I consider a better way of presenting the result.)

p.3 l. 15: instead of having the notion of weakly connected component as a footnote, please have it in the text. "Connected component" usually implies strongly, not weakly, therefore it is worth making clear what the authors are referring to in the text.

p.3 l.25 (and consequently appendix B): it is not straightforward do see the reason behind this choice of F_0 . Why this and not an expression with $((h_n-h_m)^2 - 1)$ in place of $(h_n-h_m-1)^2$ in the numerator?

p.3 eq (2.8): in the definition of trophic confusion it may be worth using x instead of h , to avoid confusion with equation (2.7).

Section 3 and 4: I believe that the manuscript would improve greatly if these two sections were swapped.

p.3 l.54: what is a basal node?

p.3 l.55: what does it mean for the network to be fully connected?

p.3 l.56: "old notions of trophic level". This ties in with my comment above about swapping the order of sections 3 and 4; Here, it would be worth summarizing a few older results and explicitly stating what are their downfalls.

Figs 1-2: what do the size of nodes and thickness of edges represent?

p.5 l.6: what does it mean for a network to be incoherent? Here the authors seem to back some known fact about IO networks with what they observe using F_0 . However, shouldn't it be the other way around, with the values of F_0 leading the authors to derive that these networks are incoherent?

p.5 l.56: What do the authors mean by "cyclic network"?

p.7 l. 10: eigenvector centrality instead of eigenvalue centrality.

p.7 l. 11: "Trophic analysis reveals that this network is strongly directional": what does directionality have to do with the value of F_0 ? The authors should try and keep their notation as consistent as possible throughout the manuscript.

p.7 l.50: "et al." instead of "et al"

p.10 ll. 37 ff: It is quite unclear the purpose of these paragraphs: please either expand further on these (by adding formulas as well, when appropriate) or remove these entirely.

Section 5: I understand what question the authors are trying to address, but it escapes me why this question should be of interest in the first place.

p. 11 ll. 53-56: "The term "normal" came from people who spent their lives with self-adjoint operators and unitary operators, both of which are normal, but people working in stability of ordinary differential equations are fully cognizant that most matrices are not normal." Please remove or rephrase this sentence.

p. 11 l 58: cut "imbalance vector" or rephrase as "implies that the imbalance vector is the zero vector: $v=0$ ".

p. 12 l. 9: "When $v = 0$ we say that the network is balanced."

p. 12 l.10: what is a normal network?

p.12 l. 17-18: The authors state the following: "if W is normal and has all eigenvalues real then $F_0 = 1$ ". Having previously noted that $F_0=1$ for symmetric matrices, the result is trivial. Indeed, a matrix is normal iff it is unitarily diagonalizable. Moreover, a unitarily diagonalizable matrix with real spectrum is Hermitian. Since the authors are assuming that W is real, "normal with all real eigenvalues" is a complicated way of saying "symmetric".

p.12 l.36: this statement (and the proof in the appendix) appears to be true only for networks without self-loops.

p.12 l.57: is $r > 1$?

p.15 l.8: "A cycle in a directed network is a closed walk in it. In contrast to some of the literature, we allow repeated edges and repeated nodes" Instead of improperly calling it cycle, the authors could refer to this object as a closed walk. Please also recall the definition of walk.

Review form: Reviewer 2

Is the manuscript scientifically sound in its present form?

Yes

Are the interpretations and conclusions justified by the results?

Yes

Is the language acceptable?

Yes

Do you have any ethical concerns with this paper?

No

Have you any concerns about statistical analyses in this paper?

No

Recommendation?

Accept as is

Comments to the Author(s)

See attached (Appendix A).

Decision letter (RSOS-201138.R0)

Dear Dr MacKay

On behalf of the Editors, I am pleased to inform you that your Manuscript RSOS-201138 entitled "How directed is a directed network?" has been accepted for publication in Royal Society Open Science subject to minor revision in accordance with the referee suggestions. Please find the referees' comments at the end of this email.

The reviewers and handling editors have recommended publication, but also suggest some minor revisions to your manuscript. Therefore, I invite you to respond to the comments and revise your manuscript.

- Ethics statement

- Data accessibility

If you wish to submit your supporting data or code to Dryad (<http://datadryad.org/>), or modify your current submission to dryad, please use the following link:
<http://datadryad.org/submit?journalID=RSOS&manu=RSOS-201138>

- Competing interests

- Authors' contributions

- Acknowledgements

- Funding statement

Because the schedule for publication is very tight, it is a condition of publication that you submit the revised version of your manuscript before 01-Aug-2020. Please note that the revision deadline will expire at 00.00am on this date. If you do not think you will be able to meet this date please let me know immediately.

If your manuscript is newly submitted and subsequently accepted for publication, you will be asked to pay the article processing charge, unless you request a waiver and this is approved by Royal Society Publishing. You can find out more about the charges at

<https://royalsocietypublishing.org/rsos/charges>. Should you have any queries, please contact openscience@royalsociety.org.

Kind regards,

Anita Kristiansen
Editorial Coordinator

on behalf of Professor Tim Rogers (Associate Editor) and Mark Chaplain (Subject Editor)
openscience@royalsociety.org

Associate Editor Comments to Author (Professor Tim Rogers):

Comments to the Author:

When making your revisions, please be sure to carefully address all of the detailed points raised by both reviewers.

In my opinion, authors should be largely free to choose their organisation and writing style, so I leave it up to you to decide if you want to take the advice of Referee 1 about these matters.

Reviewer comments to Author:

Reviewer: 1

Comments to the Author(s)

The manuscript is concerned with the introduction of a new measure of network incoherence which is based on a symmetrized graph Laplacian for weighted directed networks. The measure is tested on different real world data.

The manuscript is overall well written, but it is poorly organized and very wordy: it would benefit from reorganizing the material and incorporating in the text some of the results presented in the appendix. Moreover, many of the proofs could be carried out using formulas rather than words, and this simple switch would greatly improve readability. The comparison with other measures of incoherence should be carried out in a more thorough fashion, and it definitely deserve more space than it has been allocated in the manuscript. Disregarding directionality of edges is something that is usually best avoided, and I believe that the authors are not making a strong enough case for their decision to following this path in the manuscript.

I believe that the manuscript should be reconsidered for publication after a major revision.

There are a few points that I think are worth discussing and elaborating further on. Below I detail some of these.

p.2 eq (2.1): these quantities are extensions of the concept of "strength", introduced by Barrat et al. in [PNAS (2004), 101(11),3747-3752]. I would recommend that the authors refer to this manuscript (and subsequent generalization to digraphs, if any) and that they use the term "in/out-strength" or similar, in order to emphasize the relationship with the index introduced in 2004.

p.2, eqs (2.2)-(2.3): what is the rationale for calling these "weight" and "imbalance"?

p.2, eq (2.4): this appears to be an extension to the weighted case of the a symmetrized graph Laplacian for digraphs (as can also be seen from equation (2.5)). This type of approach to the treatment of directed graphs is often criticised in the literature, as it completely changes the topology of the network (especially in the case of highly non-symmetric matrices W). Could the authors justify their approach and explain further why disregarding directionality of edges is the right thing to do in this context? In my opinion the justification is quite weak, as it stands. (This also links to the contents of page 9).

p.3 eq (2.5): please specify who the vector u is.

p.3 l.10: here h is characterized as being *the* solution to $\lambda h = v$ (please add specification of who v is). However, in l. 14 the authors state that $\lambda h = v$ does not have a unique solution. Please change l.10 to state that h is *a* solution and fully characterize $\text{span}\{h: \lambda h = v\}$.

p.3 l.14: Instead of considering the case of a disconnected network with several weakly connected components, it would be easier to just focus on the case of weakly connected networks; Then the matrix $W+W^T$ is the (weighted) adjacency matrix of a connected undirected graph and therefore the vector of all ones spans the null space of $W+W^T$. The case of disconnected networks follows from, e.g., chapter 6.13.3 in "Networks: An Introduction" by M. Newman. (Let me clarify that I understand that the authors are doing this already, I am only suggesting what I consider a better way of presenting the result.)

p.3 l. 15: instead of having the notion of weakly connected component as a footnote, please have it in the text. "Connected component" usually implies strongly, not weakly, therefore it is worth making clear what the authors are referring to in the text.

p.3 l.25 (and consequently appendix B): it is not straightforward do see the reason behind this choice of F_0 . Why this and not an expression with $((h_n-h_m)^2 - 1)$ in place of $(h_n-h_m-1)^2$ in the numerator?

p.3 eq (2.8): in the definition of trophic confusion it may be worth using x instead of h , to avoid confusion with equation (2.7).

Section 3 and 4: I believe that the manuscript would improve greatly if these two sections were swapped.

p.3 l.54: what is a basal node?

p.3 l.55: what does it mean for the network to be fully connected?

p.3 l.56: "old notions of trophic level". This ties in with my comment above about swapping the order of sections 3 and 4; Here, it would be worth summarizing a few older results and explicitly stating what are their downfalls.

Figs 1-2: what do the size of nodes and thickness of edges represent?

p.5 l.6: what does it mean for a network to be incoherent? Here the authors seem to back some known fact about IO networks with what they observe using F_0 . However, shouldn't it be the other way around, with the values of F_0 leading the authors to derive that these networks are incoherent?

p.5 l.56: What do the authors mean by "cyclic network"?

p.7 l. 10: eigenvector centrality instead of eigenvalue centrality.

p.7 l. 11: "Trophic analysis reveals that this network is strongly directional": what does directionality have to do with the value of F_0 ? The authors should try and keep their notation as consistent as possible throughout the manuscript.

p.7 l.50: "et al." instead of "et al"

p.10 ll. 37 ff: It is quite unclear the purpose of these paragraphs: please either expand further on these (by adding formulas as well, when appropriate) or remove these entirely.

Section 5: I understand what question the authors are trying to address, but it escapes me why this question should be of interest in the first place.

p. 11 ll. 53-56: "The term "normal" came from people who spent their lives with self-adjoint operators and unitary operators, both of which are normal, but people working in stability of ordinary differential equations are fully cognizant that most matrices are not normal." Please remove or rephrase this sentence.

p. 11 l.58: cut "imbalance vector" or rephrase as "implies that the imbalance vector is the zero vector: $v=0$ ".

p. 12 l. 9: "When $v = 0$ we say that the network is balanced."

p. 12 l.10: what is a normal network?

p.12 l. 17-18: The authors state the following: "if W is normal and has all eigenvalues real then $F_0 = 1$ ". Having previously noted that $F_0=1$ for symmetric matrices, the result is trivial. Indeed, a matrix is normal iff it is unitarily diagonalizable. Moreover, a unitarily diagonalizable matrix with real spectrum is Hermitian. Since the authors are assuming that W is real, "normal with all real eigenvalues" is a complicated way of saying "symmetric".

p.12 l.36: this statement (and the proof in the appendix) appears to be true only for networks without self-loops.

p.12 l.57: is $r>1$?

p.15 l.8: "A cycle in a directed network is a closed walk in it. In contrast to some of the literature, we allow repeated edges and repeated nodes" Instead of improperly calling it cycle, the authors could refer to this object as a closed walk. Please also recall the definition of walk.

Reviewer: 2

Comments to the Author(s)

See attached (rsos.pdf)

Author's Response to Decision Letter for (RSOS-201138.R0)

See Appendix B.

Decision letter (RSOS-201138.R1)

Dear Dr MacKay,

It is a pleasure to accept your manuscript entitled "How directed is a directed network?" in its current form for publication in Royal Society Open Science. The comments of the reviewer(s) who reviewed your manuscript are included at the foot of this letter.

on behalf of Professor Tim Rogers (Associate Editor) and Mark Chaplain (Subject Editor)
openscience@royalsociety.org

Appendix A

Referee report on the manuscript
How directed is a directed network?
submitted to Royal Society Open Science

This work looks at concepts and algorithms for identifying and quantifying structure that may be hidden in pairwise interaction networks.

I found the submission to be well-written and novel, and I enjoyed reading it. The work is novel and elegant. It makes a clear contribution and is likely to have a wide impact. It combines ideas, analysis and well-chosen examples on real data sets.

I like the organization of the manuscript: getting the main point across first and discussing related work later.

I have just a couple of minor comments; these are not vital:

- It is interesting (to me) that removing the -1 in (2.7) would reduce to the classical and widely used graph Laplacian/Fielder vector structure. Perhaps this could be mentioned somewhere.
- Equations (2.7) and (2.8) are identical. I think some rewording is needed.
- The figures are generally quite compelling, however it is not always easy to see all the edges and to identify their direction. Figure 4 is the most extreme example. Is there any way of dealing with this?
- The Discussion section undersells the material and finishes on a strange note. Given that many readers will go straight there, I would recommend a longer and more forceful description of the contributions.

Appendix B

Response to reviewers' comments

How directed is a directed network?
submitted to Royal Society Open Science

Contents

1	Response to Associate Editor.....	1
2	Response to Reviewer 1.....	1
3	Response to Reviewer 2:	8

Key

- Referees comments in black
- Our responses in red

1 Response to Associate Editor

Associate Editor Comments to Author (Professor Tim Rogers):

Comments to the Author:

When making your revisions, please be sure to carefully address all of the detailed points raised by both reviewers.

In my opinion, authors should be largely free to choose their organisation and writing style, so I leave it up to you to decide if you want to take the advice of Referee 1 about these matters.

We are grateful for the reviewers' comments and for the freedom you are giving to us to decide about organisation of the paper and writing style. We chose the organisation deliberately: after an introduction which mentions the ways in which we've improved over previous methods, we present our method and give some illustrations; then we make a comparison with previous methods, followed by several significant connections to other network properties. We also chose to relegate most of the mathematics to appendices because we wanted to keep the paper accessible to less mathematically oriented readers, especially from social science, where we believe the paper can have big impact. We wish to keep to this organisation. On writing style, we feel the style we have adopted is appropriate; again, we chose it to attempt to keep on board readers of a less mathematically oriented background, so it is perforce more wordy than some papers.

2 Response to Reviewer 1

Reviewer comments to Author:

Reviewer: 1

Comments to the Author(s)

The manuscript is concerned with the introduction of a new measure of network incoherence which is based on a symmetrized graph Laplacian for weighted directed networks. The measure is tested on different real world data.

The manuscript is overall well written, but it is poorly organized and very wordy: it would benefit from reorganizing the material and incorporating in the text some of the results presented in the appendix.

We chose the organisation of the material deliberately, to present the method as early as possible, illustrate its use to attract the general reader's interest, and then discuss in detail comparisons with previous methods, followed by connecting to other network properties. We also chose deliberately to put most of the mathematical proofs into appendices, so that less mathematically inclined readers would not be put off, because we believe a major domain of impact for the method will be the social sciences. Furthermore, the other reviewer liked the organisation!

Moreover, many of the proofs could be carried out using formulas rather than words, and this simple switch would greatly improve readability.

Most of the proofs are done by formulae. The other proofs are written so that a less mathematically inclined reader can follow them.

The comparison with other measures of incoherence should be carried out in a more thorough fashion, and it definitely deserve more space that it has been allocated in the manuscript.

There is only one established other measure of incoherence of which we are aware and that is the one of [JDDM] which we cover thoroughly. We have expanded our comments on the notion used by [CHK]. We consider our comments on the notions in [T] and [LM] sufficient. We make the connection with 'circularity' of [KIII].

Disregarding directionality of edges is something that is usually best avoided, and I believe that the authors are not making a strong enough case for their decision to following this path in the manuscript.

It is a misunderstanding to say we have disregarded the directionality of edges. The whole paper is about directed networks. Although the graph-Laplacian is symmetric, the imbalance vector is antisymmetric and that is where the directionality is encoded.

I believe that the manuscript should be reconsidered for publication after a major revision.

There are a few points that I think are worth discussing and elaborating further on. Below I detail some of these.

[Detailed points start here¹]

1. p.2 eq (2.1): these quantities are extenstios of the cocept of "strength", introduced by Barrat et al. in [PNAS (2004), 101(11),3747-3752]. I would

¹ We have numbered these to help with discussion/for reference, but they were not numbered by referee.

recommend that the authors refer to this manuscript (and subsequent generalization to digraphs, if any) and that they use the term "in/out-strength" or similar, in order to emphasize the relationship with the index introduced in 2004. We have added a reference to that paper, but do not consider that we have to use its terminology. Furthermore, "strength" in that paper refers only to the out-weight.

2. p.2, eqs (2.2)-(2.3): what is the rationale for calling these "weight" and "imbalance"? We have added the (obvious in our eyes) rationale for calling v the imbalance. We consider it self-evident why we call u the weight of the node, because it is the sum of the associated edge weights in and out of the node, but to distinguish it we've inserted (total) in front of weight, though this differs from Barrat et al.'s use of the term.
3. p.2. eq (2.4): this appears to be an extension to the weighted case of the a symmetrized graph Laplacian for digraphs (as can also be seen from equation (2.5)). This type of approach to the treatment of directed graphs is often criticised in the literature, as it completely changes the topology of the network (especially in the case of highly non-symmetric matrices W). Could the authors justify their approach and explain further why disregarding directionality of edges is the right thing to do in this context? In my opinion the justification is quite weak, as it stands. (This also links to the contents of page 9). The asymmetry of the network is encapsulated in the imbalance vector.
4. p.3 eq (2.5): please specify who the vector u is. This was defined in (2.3), but we have now spelt out that we make vector u from the u_n .
5. p.3 l.10: here h is characterized as being *the* solution to $\Lambda h = v$ (please add specification of who v is). However, in l. 14 the authors state that $\Lambda h = v$ does not have a unique solution. Please change l.10 to state that h is *a* solution and fully characterize $\text{span}\{h: \Lambda h = v\}$. We have added "modulo shifts".
6. p.3 l.14: Instead of considering the case of a disconnected network with several weakly connected components, it would be easier to just focus on the case of weakly connected networks; Then the matrix $W+W^T$ is the (weighted) adjacency matrix of a connected undirected graph and therefore the vector of all ones spans the null space of $W+W^T$. The case of disconnected networks follows from, e.g., chapter 6.13.3 in "Networks: An Introduction" by M. Newman. (Let me clarify that I understand that the authors are doing this already, I am only suggesting what I consider a better way of presenting the result.) Thank you for the suggestion, but we prefer the way we have presented it.
7. p.3 l. 15: instead of having the notion of weakly connected component as a footnote, please have it in the text. "Connected component" usually implies strongly, not weakly, therefore it is worth making clear what the

authors are referring to in the text. **OK, we've moved the footnote into the text.**

8. p.3 l.25 (and consequently appendix B): it is not straightforward do see the reason behind this choice of F_0 . Why this and not an expression with $((h_n-h_m)^2 - 1)$ in place of $(h_n-h_m-1)^2$ in the numerator? **We motivated the choice in the next paragraph.**
9. p.3 eq (2.8): in the definition of trophic confusion it may be worth using x instead of h , to avoid confusion with equation (2.7). **We do not consider it confusing, but have added some text to clarify.**
10. Section 3 and 4: I believe that the manuscript would improve greatly if these two sections were swapped. **We consider it better to illustrate the method before comparing it with previous ones. We admit that readers who already know the previous ones might prefer the comparison before the illustrations, but we are targeting a broad audience who may not have come across them before and we want to get the value across first.**
11. p.3 l.54: what is a basal node? **We defined it on p.2, but have amplified the definition there to make it clearer.**
12. p.3 l.55: what does it mean for the network to be fully connected? **We have changed this to completely connected, as perhaps the more standard term.**
13. p.3 l.56: "old notions of trophic level". This ties in with my comment above about swapping the order of sections 3 and 4; Here, it would be worth summarizing a few older results and explicitly stating what are their downfalls. **We did this in the Introduction!**
14. Figs 1-2: what do the size of nodes and thickness of edges represent? **In fig.1 the nodes are all the same size and the edges are all the same thickness; this corresponds to the edge weights all being 1, as stated in the caption. In fig.2 the caption explains the node sizes and edge thicknesses.**
15. p.5 l.6: what does it mean for a network to be incoherent? Here the authors seem to back some known fact about IO networks with what they observe using F_0 . However, shouldn't it be the other way around, with the values of F_0 leading the authors to derive that these networks are incoherent? **We mean that F_0 is not small. In case this was not clear, we have modified the text a bit to emphasise it.**
16. p.5 l.56: What do he authors mean by "cyclic network"? **A network with some cycles.**
17. p.7 l. 10: eigenvector centrality instead of eigenvalue centrality. **Yes, thanks.**

18. p.7 l. 11: "Trophic analysis reveals that this network is strongly directional": what does directionality have to do with the value of F_0 ? The authors should try and keep their notation as consistent as possible throughout the manuscript. **Yes, we had neglected to make this connection. We've added text to the end of Sec.2 to explain it.**
19. p.7 l.50: "et al." instead of "et al" **OK**
20. p.10 ll. 37 ff: It is quite unclear the purpose of these paragraphs: please either expand further on these (by adding formulas as well, when appropriate) or remove these entirely. **They are in response to a previous reviewer, who insisted we cite some literature on hierarchical analysis of networks and that we relate to centrality measures. We consider it best to leave them as they stand.**
21. Section 5: I understand what question the authors are trying to address, but it escapes me why this question should be of interest in the first place. **Good point. We have added an explanation of the interest.**
22. p. 11 ll. 53-56: "The term "normal" came from people who spent their lives with self-adjoint operators and unitary operators, both of which are normal, but people working in stability of ordinary differential equations are fully cognizant that most matrices are not normal." Please remove or rephrase this sentence. **OK, we've replaced this with something less critical, but it is essential in our opinion to explain to less mathematical readers that there is nothing "normal" about normal operators!**
23. p. 11 l 58: cut "imbalance vector" or rephrase as "implies that the imbalance vector is the zero vector: $v=0$ ". **OK, rewritten.**
24. p. 12 l. 9: "When $v = 0$ we say that the network is balanced." **Not clear what the question or comment is here, but we've rephrased the next sentence to hopefully make its status clearer.**
25. p. 12 l.10: what is a normal network? **We defined it in (6.1)!**
26. p.12 l. 17-18: The authors state the following: "if W is normal and has all eigenvalues real then $F_0 = 1$ ". Having previously noted that $F_0=1$ for symmetric matrices, the result is trivial. Indeed, a matrix is normal iff it is unitarily diagonalizable. Moreover, a unitarily diagonalizable matrix with real spectrum is Hermitian. Since the authors are assuming that W is real, **"normal with all real eigenvalues" is a complicated way of saying "symmetric". Good point! We've deleted it and the associated appendix. Many thanks!**
27. p.12 l.36: this statement (and the proof in the appendix) appears to be true only for networks without self-loops. **It is trivially true for networks with self-loops: they are never maximally coherent, because the height**

difference for a self-loop is 0 whereas to be maximally coherent it has to be 1.

28. p.12 l.57: is $r > 1$? Not necessarily, so we've changed "reduction" to "scaling".
29. p.15 l.8: "A cycle in a directed network is a closed walk in it. In contrast to some of the literature, we allow repeated edges and repeated nodes" Instead of improperly calling it cycle, the authors could refer to this object as a closed walk. Please also recall the definition of walk. We prefer to call it a cycle and explain that we're using it in a more general sense than graph theorists. This is because we use the word many times and it is shorter than "closed walk", furthermore it is a more intuitive term for people with less mathematical backgrounds. But we agree we should define walks and closed walks, so we have added them.

3 Response to Reviewer 2:

Reviewer comments to Author: Reviewer: 2

Referee report on the manuscript
How directed is a directed network?
submitted to Royal Society Open Science

This work looks at concepts and algorithms for identifying and quantifying structure that may be hidden in pairwise interaction networks. I found the submission to be well-written and novel, and I enjoyed reading it. The work is novel and elegant. It makes a clear contribution and is likely to have a wide impact. It combines ideas, analysis and well-chosen examples on real data sets.

I like the organization of the manuscript: getting the main point across first and discussing related work later.

I have just a couple of minor comments; these are not vital:

[Detailed comments start here]

1. It is interesting (to me) that removing the -1 in (2.7) would reduce to the classical and widely used graph Laplacian/Fiedler vector structure. Perhaps this could be mentioned somewhere. Removing the -1 would lead to $\lambda h = 0$, which is only the equation for constants on connected components. The Fiedler vector is the solution (up to scale) of $\lambda h = \lambda_2 h$ with λ_2 the second eigenvalue. That is something very different from what we are solving. We decided not to mention it.

2. Equations (2.7) and (2.8) are identical. I think some rewording is needed. The right-hand sides are indeed the same. We've added text that the first occurrence is for the trophic levels, while the second one is a variational principle to determine them.
3. The figures are generally quite compelling, however it is not always easy to see all the edges and to identify their direction. Figure 4 is the most extreme example. Is there any way of dealing with this? Fig.4 makes clear that there is a strong directionality in the network.
4. The Discussion section undersells the material and finishes on a strange note. Given that many readers will go straight there, I would recommend a longer and more forceful description of the contributions. Agreed on both counts. We've rewritten it and moved the extension to arbitrary target height differences to a new section, immediately preceding. We also moved the material from the corresponding appendix to the new section.